# The *wtf* meiotic driver gene family has unexpectedly persisted for over 100 million years

Mickaël De Carvalho[1,2†], Guo-Song Jia[3,4†], Ananya Nidamangala Srinivasa[1,5], R Blake Billmyre[1], Yan-Hui Xu[4], Jeffrey J Lange[1], Ibrahim M Sabbarini[1], Li-Lin Du[4,6]*, Sarah E Zanders[1,5]*

[1]Stowers Institute for Medical Research, Kansas City, United States; [2]Open University, Milton Keynes, United Kingdom; [3]PTN Joint Graduate Program, School of Life Sciences, Tsinghua University, Beijing, China; [4]National Institute of Biological Sciences, Beijing, Beijing, China; [5]Department of Molecular and Integrative Physiology, University of Kansas Medical Center, Kansas City, United States; [6]Tsinghua Institute of Multidisciplinary Biomedical Research, Tsinghua University, Beijing, China

**\*For correspondence:**
dulilin@nibs.ac.cn (LLD);
sez@stowers.org (SEZ)

[†]These authors contributed equally to this work

**Competing interest:** The authors declare that no competing interests exist.

**Abstract** Meiotic drivers are selfish elements that bias their own transmission into more than half of the viable progeny produced by a driver+/driver− heterozygote. Meiotic drivers are thought to exist for relatively short evolutionary timespans because a driver gene or gene family is often found in a single species or in a group of very closely related species. Additionally, drivers are generally considered doomed to extinction when they spread to fixation or when suppressors arise. In this study, we examine the evolutionary history of the *wtf* meiotic drivers first discovered in the fission yeast *Schizosaccharomyces pombe*. We identify homologous genes in three other fission yeast species, *S. octosporus*, *S. osmophilus*, and *S. cryophilus*, which are estimated to have diverged over 100 million years ago from the *S. pombe* lineage. Synteny evidence supports that *wtf* genes were present in the common ancestor of these four species. Moreover, the ancestral genes were likely drivers as *wtf* genes in *S. octosporus* cause meiotic drive. Our findings indicate that meiotic drive systems can be maintained for long evolutionary timespans.

## Editor's evaluation

This paper presents important findings on the long-term evolutionary persistence of a meiotic driver gene family across several species of fission yeasts. The authors provide compelling evidence from phylogenetic analyses, comparative genomics, and functional experiments that *wtf* genes have an ancient origin in *Schizosaccharomyces* and retain the ability to drive. Based on their finding of extensive gene duplication and gene conversion throughout the evolutionary history of *wtf* genes, the authors also present an interesting hypothesis to explain how the ability to drive might be maintained over long evolutionary timescales – namely, that it is a property of the gene family as a whole, rather than of a single locus. Whether such a scenario is unique to fission yeasts or applies more broadly to other taxa is currently unknown, but this work certainly represents one of the most detailed mechanistic studies of selfish genes in any wild species to date, and thus provides a valuable example for future studies of how such systems might evolve.

## Introduction

During meiosis, the two alleles at a given locus segregate from each other and are each transmitted into an equal number of the viable gametes produced by a heterozygous organism. This fundamental rule of genetics is known as Mendel's law of segregation (*Abbott and Fairbanks, 2016*). Most genetic loci follow this law, which facilitates natural selection by allowing alternate variants to compete on an even playing field (*Crow, 1991*). Meiotic drivers, however, are genetic loci that manipulate gametogenesis to gain an unfair transmission advantage into gametes. Rather than being transmitted to 50% of the gametes produced by a driver+/driver− heterozygote, meiotic drivers are transmitted to most or even all of the functional gametes (*Sandler and Novitski, 1957*; *Zimmering et al., 1970*).

Meiotic drivers are found in diverse eukaryotes including plants, fungi, and animals (*Bravo Núñez et al., 2018b*; *Burt and Trivers, 2006*; *Courret et al., 2019a*; *Lindholm et al., 2016*). Despite their broad phylogenetic distribution, drivers in different systems are not thought to share common evolutionary origins. Instead, empirical observations combined with theoretical work have led to the expectation that drivers are evolutionarily short-lived (*Burt and Trivers, 2006*). Specifically, drivers are believed to have been born repeatedly, but each driver can only persist for a short evolutionary period before extinction, and as a result, drive systems are lineage-specific (*Hatcher, 2000*; *Price et al., 2019*).

Understanding the birth of a driver is conceptually straightforward: if a sequence acquires the ability to drive, it can spread in the population (*Crow, 1991*). The paths to driver extinction are more complex, but one route to extinction is through suppression (*Bastide et al., 2011*; *Bravo Núñez et al., 2018a*; *Carvalho and Vaz, 1999*; *Courret et al., 2019b*; *Tao et al., 2007*; *Unckless et al., 2015*). Drive is generally costly to fitness. The cost of drive can result directly from the drive mechanism. For example, some drivers act by destroying gametes that do not inherit them (*Bravo Núñez et al., 2018b*). Drivers can also decrease fitness indirectly through many mechanisms, including by disrupting Mendelian allele transmission (*Zanders and Unckless, 2019*). Because of these costs, natural selection is thought to favor the evolution of drive suppressors (*Cazemajor et al., 1997*; *Crow, 1991*; *Finseth et al., 2021*; *Kumon et al., 2021*; *Veller, 2022*). Suppressed drivers have no transmission advantage and are expected to accumulate inactivating mutations (*Burt and Trivers, 2006*). In a second path to driver extinction, the driver evades suppression and spreads to fixation. If the driver is on a sex chromosome or the driving haplotype acquires strongly deleterious mutations, driver fixation can lead to driver extinction via host extinction (*Dyer et al., 2007*; *Hamilton, 1967*). If the fixed driver is autosomal, it experiences no transmission advantage and can accumulate inactivating mutations, in a fate similar to that of suppressed drivers.

The molecularly identified meiotic drivers largely support the idea that drivers have limited evolutionary lifespans and confined species distributions, with a driver gene or gene family often only found in a single species (*Finseth et al., 2021*; *Lindholm et al., 2016*; *Lyon, 2003*; *Price et al., 2019*; *Zanders and Johannesson, 2021*). In *Drosophila*, for example, the sister species *Drosophila melanogaster* and *D. simulans* shared a common ancestor 5.4 million years ago (*Tamura et al., 2003*), but they each contain distinct meiotic drive systems (*Cazemajor et al., 1997*; *Helleu et al., 2016*; *Larracuente and Presgraves, 2012*; *Lin et al., 2018*; *Tao et al., 2007*).

There are a few known exceptions where a drive gene is found in more than one species. For example, sequences homologous to the *Dox* driver of *D. simulans* are also found in *D. mauritiana* and *D. sechellia* (*Muirhead and Presgraves, 2021*; *Vedanayagam et al., 2021*). Although there have been more recent introgressions involving *Dox* between *D. simulans* and *D. mauritiana*, sequences homologous to *Dox* appear to have existed 0.2 million years ago in the ancestor of the *D. simulans* clade (*Meiklejohn et al., 2018*; *Muirhead and Presgraves, 2021*; *Vedanayagam et al., 2021*). In rice (*Oryza*), many meiotic drive systems and potential meiotic drive loci have been mapped as sterility loci in crosses between domesticated varieties/species, or between domesticated and wild varieties/species (representing up to ~0.9 million years of divergence). Homologs of genes in these drive systems exist in more distantly related rice species, but whether they are meiotic drivers or precursors of drivers is unclear (*Chen et al., 2008*; *Huang et al., 2015*; *Koide et al., 2018*; *Long et al., 2008*; *Sakata et al., 2021*; *Shen et al., 2017*; *Xie et al., 2019*; *Xie et al., 2017*; *Yang et al., 2012*; *Yu et al., 2016*). Another crop drive system is the 'knobs' found in maize (*Zea mays*) and its wild relative *Tripsacum dactyloides*. These two species diverged about 1 million years ago (*Ross-Ibarra et al., 2009*), but drive of knobs has only been conclusively demonstrated in maize (*Dawe et al., 2018*; *Kanizay*

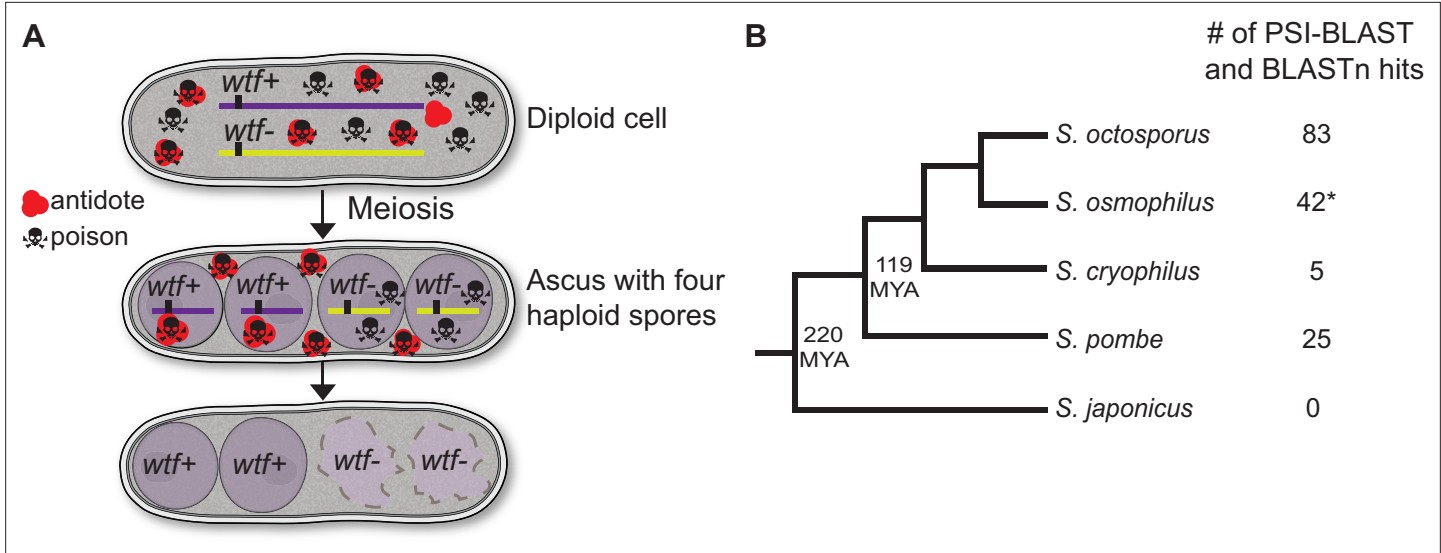

**Figure 1.** *wtf* homologs are found outside of *Schizosaccharomyces pombe*. (**A**) Model for meiotic drive of *wtf* genes in *S. pombe*, modified from ***Nuckolls et al., 2017***. All spores are exposed to the poison protein, but those that inherit the *wtf* driver are rescued by the antidote protein. (**B**) Schematic phylogeny of *Schizosaccharomyces* species based on published reports (***Brysch-Herzberg et al., 2019***; ***Rhind et al., 2011***). Our analyses of percent identity between orthologs (***Supplementary file 1a***) agree with this tree topology. MYA represents million years ago. Annotations of all the identified *S. osmophilus* genes can be found in ***Figure 1—source data 2***. To the right of the phylogeny, we list the numbers of *wtf* homologs found by position-specific iterated-basic local alignment search tool (PSI-BLAST) and BLASTn searches. *The *S. osmophilus* genome is not fully assembled, so the number represents the *wtf* homologs found within the assembled contigs.

The online version of this article includes the following source data and figure supplement(s) for figure 1:

**Source data 1.** *Schizosaccharomyces osmophilus* genome assembly.

**Source data 2.** Predicted *Schizosaccharomyces osmophilus* gene annotations.

**Source data 3.** Orthologous genes in *Schizosaccharomyces*.

**Source data 4.** *Schizosaccharomyces octosporus* genome annotation.

**Source data 5.** *Schizosaccharomyces osmophilus* genome annotation.

**Source data 6.** *Schizosaccharomyces cryophilus* genome annotation.

**Figure supplement 1.** Maps of the *wtf* gene family members in *Schizosaccharomyces octosporus, S. osmophilus, S. cryophilus,* and *S. pombe.*

*et al., 2013*). In fungi, the *Spok* genes first identified as drivers in *Podospora anserina* are found in several distantly related fungal species. However, horizontal gene transfer is a more likely explanation for the phylogenetic distribution of *Spok* genes than shared decent from a common ancestor (***Grognet et al., 2014***; ***Vogan et al., 2021***).

In this work, we explore the long-term evolutionary dynamics of drive systems using the recently discovered *wtf* drivers as a model system. *wtf* driver genes are found in all sequenced isolates of the fission yeast *S. pombe*. Each sequenced *S. pombe* isolate contains between 4 and 14 distinct predicted *wtf* driver genes (***Eickbush et al., 2019***; ***Hu et al., 2017***). The *wtf* drivers are killer meiotic drivers as they act by destroying the meiotic products (spores) that do not inherit the *wtf* driver from a *wtf+/wtf−* heterozygote (***Figure 1A***). To cause selective spore death, each *wtf* drive gene employs a poison protein and an antidote protein produced from two overlapping transcripts. All developing spores are exposed to the poison, while only spores that inherit the *wtf* driver gene express the antidote and are rescued from destruction (***Bravo Núñez et al., 2018a***; ***Bravo Núñez et al., 2020a***; ***Bravo Núñez et al., 2020b***; ***Hu et al., 2017***; ***Nuckolls et al., 2017***).

Here, we analyzed the phylogenetic distribution of *wtf* genes and found highly diverged but homologous *wtf* genes in *S. octosporus, S. osmophilus,* and *S. cryophilus,* three species that diverged more than 100 million years ago from the *S. pombe* lineage (***Brysch-Herzberg et al., 2019***; ***Rhind et al., 2011***). Analyses of synteny support that the *wtf* gene family existed in the common ancestor of *S. pombe* and these three other fission yeast species. Like the *S. pombe wtf* genes, the newly described *wtf* genes exhibit evolutionary signatures of genetic conflict, namely rapid evolution. Moreover, we

demonstrate that at least one *wtf* gene in each of the species can encode poison and antidote proteins on overlapping coding sequences. We investigated a subset of the *S. octosporus wtf* genes further and found that some cause meiotic drive when heterozygous. We conclude that *wtf* drivers have maintained the capacity to drive for over 100 million years. Finally, we speculate that the *wtf* drivers outrun extinction through perpetual gene birth and renewal via recombination mechanisms.

## Results

### Genes with homology to *wtf* drivers are found in *S. octosporus, S. osmophilus,* and *S. cryophilus*

As a first step in understanding the long-term evolution of the *wtf* meiotic drivers, we analyzed the phylogenetic distribution of the *wtf* gene family. There are four described *Schizosaccharomyces* species in addition to *S. pombe: S. octosporus, S. osmophilus, S. cryophilus,* and *S. japonicus* (***Figure 1B***; ***Brysch-Herzberg et al., 2019***; ***Rhind et al., 2011***). These species are thought to have shared a common ancestor around 200 million years ago. The amino acid divergence of 1:1 orthologs between *S. pombe* and *S. japonicus* is 55%, which is similar to that observed between humans and lancelets (a cephalochordate) (***Rhind et al., 2011***).

At the time this work was initiated, genome assemblies were available for all species except *S. osmophilus*, which was only recently described (***Brysch-Herzberg et al., 2019***). We therefore sequenced the genome of the *S. osmophilus* strain CBS 15792 using both Illumina paired-end reads and Oxford nanopore reads. We used these data to assemble a draft of the *S. osmophilus* genome consisting of 11 contigs. We then predicted the coding sequences of *S. osmophilus* genes using *S. octosporus* annotations as a guide (***Figure 1—source data 2***; ***Tong et al., 2019***; ***Hoff and Stanke, 2018***). We found that 1:1 orthologs between *S. osmophilus* and *S. octosporus* share 88.9% amino acid identity on average, while *S. osmophilus* and *S. cryophilus* orthologs share 85.2% amino acid identity on average (***Xu et al., 2019***; ***Supplementary file 1a***). Our results are consistent with the previously proposed phylogeny of the fission yeasts that used limited sequencing data from *S. osmophilus* (***Brysch-Herzberg et al., 2019***; ***Figure 1B***).

We next searched for *wtf* gene homologs in the genome assemblies of *S. octosporus, S. osmophilus, S. cryophilus,* and *S. japonicus*. Even within *S. pombe*, the *wtf* genes are diverse, and a standard BLAST (basic local alignment search tool) search using one *wtf* gene as a query does not identify all members of the family (***Altschul et al., 1990***). Because of this, we searched for homologs using PSI-BLAST (position-specific iterated BLAST). PSI-BLAST uses the results from an initial search to create a profile of the multi-alignment between the query protein and the best hits. This profile is then used to find other proteins, and the iterative process continues until no more significant hits are found (***Altschul et al., 1997***). Using the protein encoded by *S. pombe wtf4* as an initial query, we were able to find potential *wtf* homologs in *S. octosporus, S. osmophilus,* and *S. cryophilus* but not *S. japonicus* (***Figure 1B***). We repeated PSI-BLAST searches using as queries proteins encoded by candidate *wtf* genes from non-*pombe* species (*S. octosporus wtf25, S. cryophilus wtf1,* and *S. osmophilus wtf14*). These searches all identified *S. pombe* Wtf proteins as hits. None of our PSI-BLAST searches found candidate *wtf* homologs in *S. japonicus* or outside of fission yeasts.

We then used the nucleotide sequences of candidate *wtf* genes as queries to perform additional BLASTn searches to find potential pseudogenes missed by our PSI-BLAST searches. For example, we used the nucleotide sequences of all the *S. octosporus wtf* genes identified by the PSI-BLAST search as queries to search for homologous pseudogenes within *S. octosporus*. Only hits more than 200 base pairs long were considered, although there were additional shorter hits that are likely homologous. We then used sequence alignments of candidate *wtf* genes within each species, and sometimes between species, to refine the predicted coding sequences. In *S. octosporus*, we also generated Oxford nanopore long-read RNA sequencing data (NCBI SRA SRR17543072 and SRR17543073) from a meiotic sample and used it to facilitate the delineation of exon-intron boundaries of *wtf* genes.

Overall, we identified 48 predicted *wtf* genes and 35 predicted *wtf* pseudogenes in *S. octosporus*; 31 predicted *wtf* genes and 11 predicted *wtf* pseudogenes in *S. osmophilus;* and 2 predicted *wtf* genes and 3 predicted *wtf* pseudogenes in *S. cryophilus* (***Figure 1B***; ***Figure 1—figure supplement 1***; ***Supplementary file 1b-d***). Previously, 16 intact *wtf* genes and 9 pseudogenes were described in the reference isolate of *S. pombe* (***Bowen et al., 2003***; ***Eickbush et al., 2019***; ***Hu et al., 2017***). We were

concerned that the lack of PSI-BLAST hits in *S. japonicus* could have been due to extensive divergence rather than a lack of potential *wtf* gene homologs. However, a more extensive search not dependent on high sequence homology also failed to find potential *wtf* homologs in *S. japonicus* (see Methods).

## Candidate *wtf* genes of *S. octosporus*, *S. osmophilus*, and *S. cryophilus* share additional features with *S. pombe wtf* genes

The homology between the *S. pombe wtf* genes and those found in the other *Schizosaccharomyces* species is low (*Figure 2—figure supplement 1*). For example, the most similar *wtf* gene pair between *S. pombe* (FY29033 *wtf25*) and *S. octosporus* (*S. octosporus wtf56*) shares only 16% amino acid identity, compared to an average of 65.3% amino acid identity between orthologous gene pairs (*Supplementary file 1a and e*). Given this high divergence, we examined features other than protein sequences to further test if the candidate *wtf* genes are truly members of the *wtf* gene family.

We first looked for similarities in overall gene structure between the *S. pombe wtf* genes and the candidate *wtf* genes in *S. octosporus*, *S. osmophilus*, and *S. cryophilus*. The *wtf* genes of *S. pombe* have been classified into three broad categories. The first two categories have been functionally characterized and include predicted meiotic drivers (4–14 per isolate) and predicted suppressors of drive that encode only antidote proteins (9–17 per isolate) (*Bravo Núñez et al., 2018a*; *Bravo Núñez et al., 2020b*; *Eickbush et al., 2019*; *Hu et al., 2017*; *Nuckolls et al., 2017*). The final class of *wtf* genes is comprised of four genes that share some structural and expression patterns with *wtf* drivers and suppressors but have no demonstrated drive phenotypes (*Bravo Núñez et al., 2020a*; *Eickbush et al., 2019*). These four *wtf* genes are also quite diverged from each other and all other *wtf* genes, with each unknown gene forming a distinct clade in a phylogeny of *S. pombe wtf* genes (*Eickbush et al., 2019*). We found that the overall gene structure of the candidate *wtf* genes in *S. octosporus*, *S. osmophilus*, and *S. cryophilus* was similar to the 5-exon *wtf* drivers and 5-exon *wtf* suppressors in *S. pombe* (*Figure 2A*). Moreover, the relative sizes of the corresponding exons and introns are remarkably similar between the species, even though the actual sequences are generally quite different (*Figure 2—figure supplement 1* and *Figure 2—source data 1*).

We next looked for similarities between promoters controlling the transcription of the *S. pombe wtf* genes and the potential promoters of the candidate *wtf* genes in other species. The promoters of the *S. pombe wtf4* gene are representative of the promoters of *wtf* drivers in *S. pombe* (*Nuckolls et al., 2021*). The Wtf4$^{antidote}$ protein is encoded on exons 1–6, with the promoter found upstream of exon 1. We found no shared homology between the *S. pombe wtf*$^{antidote}$ promoter sequences and sequences upstream of exon 1 in the candidate *wtf* genes found in the other species.

The Wtf4$^{poison}$ protein is encoded on exons 2–6, and the promoter is found within what is intron 1 of the *wtf4*$^{antidote}$ transcript. The *S. pombe wtf4* poison promoter contains a cis-regulatory FLEX motif that is bound by the Mei4 master meiotic transcription factor and is essential for expression of the Wtf4$^{poison}$ protein (*Nuckolls et al., 2021*). The consensus sequence of the FLEX motif has been defined as GTAAACAAACA(A/T)A(A/C), with the first 11 nucleotides being more invariant (*Abe and Shimoda, 2001*). All verified *S. pombe wtf* drivers contain in their intron 1 the 11 bp GTAAACAAACA FLEX motif sequence (*Nuckolls et al., 2021*).

To examine whether Mei4 also regulates the expression of the candidate *wtf* genes outside of *S. pombe*, we first analyzed the conservation of the Mei4-binding motif. To do this, we first compiled a list of 49 *S. pombe* Mei4 target genes that have 1:1:1:1 orthologs in *S. octosporus*, *S. osmophilus*, and *S. cryophilus* (*Supplementary file 1f*) and used MEME (multiple em for motif elicitation) to perform de novo motif discovery in 1000 bp sequences upstream of the start codons of this set of genes in each species (*Bailey et al., 2015*). Manual inspection of the MEME-discovered motifs revealed that the FLEX motif is highly conserved in these four species (*Figure 2—figure supplement 2A*). Combining the 196 genes from all four species as the input for MEME analysis resulted in a 11 bp motif matching the GTAAACAAACA FLEX motif sequence (*Figure 2—figure supplement 2A*). This MEME-identified 11 bp motif was submitted to the FIMO (find individual motif occurrences) tool of the MEME suite to perform motif scanning in the genomes of the four species using the default p-value cutoff of 1E-4. We then compared the number of FIMO hits in known Mei4 targets in *S. pombe* to number of hits in other *S. pombe* genes (not thought to be Mei4 targets) when using the default cutoff and when using more conservative p-value cutoffs. We found that a p-value cutoff of 3E-6 worked well for distinguishing Mei4 targets from non-targets. We therefore sorted the 33089 FIMO hits into unreliable hits

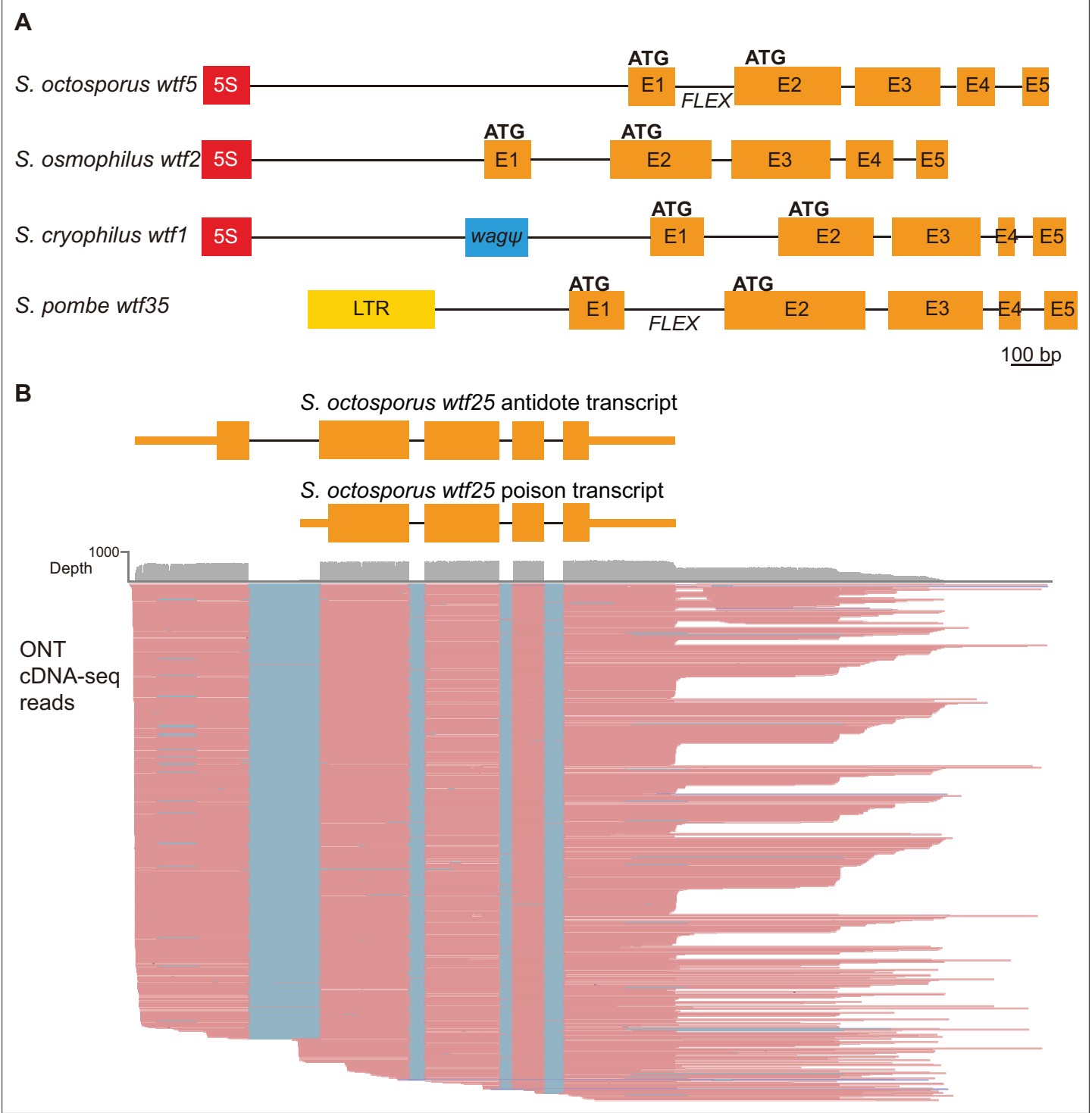

**Figure 2.** *Schizosaccharomyces pombe wtf* genes share features with other *wtf* genes outside of *S. pombe*. (**A**) Schematic *wtf* loci of the *Schizosaccharomyces* species. Orange boxes correspond to exons (E1 indicates exon 1, E2 indicates exon 2, etc.), the red boxes represent 5S rDNA genes, the blue box represents a pseudogenized *wag* gene, and the yellow box is a long terminal repeat (LTR) from a Tf transposon. The predicted translational start sites for the antidote (ATG in exon 1) and poison (ATG in exon 2) proteins are indicated, as is the FLEX transcriptional regulatory motif (***Supplementary file 1b-d***). (**B**) Long-read RNA sequencing of mRNAs from meiotic *S. octosporus* cells revealed two main transcript isoforms of the *wtf25* gene, presumably encoding an antidote and a poison protein, respectively. cDNA reads obtained using the Oxford Nanopore Technologies (ONT) platform are shown in pink. Blue lines indicate sequences missing in the reads due to splicing. The diagram at the top depicts the two main transcript isoforms. The 3' transcript ends shown in the diagram correspond to the major transcript end revealed by cDNA reads.

*Figure 2 continued on next page*

*Figure 2 continued*

The online version of this article includes the following source data and figure supplement(s) for figure 2:

**Source data 1.** wtf genes share similar exon and intron sizes.

**Figure supplement 1.** Limited conservation of Wtf proteins.

**Figure supplement 1—source data 1.** Multi-alignment of all 113 Wtf predicted antidote proteins of *Schizosaccharomyces octosporus*, *S. osmophilus*, *S. cryophilus*, and *S. pombe*.

**Figure supplement 2.** Many *wtf* genes in *Schizosaccharomyces octosporus* harbor the FLEX motif in intron 1.

**Figure supplement 3.** Transcription levels of predicted poison and antidote isoforms of intact *wtf* genes in *Schizosaccharomyces octosporus*.

(p-value >3E-6) and confident hits (p-value ≤ 3E-6). A total of 2917 confident hits (476, 716, 827, and 898 in *S. pombe*, *S. octosporus*, *S. osmophilus,* and *S. cryophilus*, respectively) were found (***Supplementary file 1g-h***).

As expected, among the *wtf* genes in the *S. pombe* reference genome, only the four genes that can express the *wtf^poison* transcript (*wtf4*, *wtf13*, *wtf19*, and *wtf23*) possess confident FIMO hits in intron 1 (***Figure 2—figure supplement 2B***). Inspecting intron 1 of the candidate *wtf* genes in the other three species showed that 20 of the 48 intact *wtf* candidate genes in *S. octosporus* possess confident FIMO hits in intron 1, 1 of the 2 intact *wtf* candidate genes in *S. cryophilus* possesses confident FIMO hits in intron 1, whereas none of the 31 intact *wtf* candidate genes in *S. osmophilus* possesses confident FIMO hits in intron 1 (***Figure 2—figure supplement 2B***, ***Supplementary file 1b-d***). Thus, the presence of the FLEX motif in intron 1 (defined by confident FIMO hit) appears to be a feature conserved in 42% of candidate *wtf* genes in *S. octosporus*.

To assess whether the presence of the FLEX motif in intron 1 of *wtf* candidate genes in *S. octosporus* is an indication of the ability to express the *wtf^poison* transcript, we analyzed our long-read RNA sequencing data of meiotic *S. octosporus* cells. All 48 intact *S. octosporus wtf* candidate genes have long transcripts initiating from upstream of exon 1, and 31 of them also have detectable short transcripts initiating from within intron 1 (***Figure 2B***, ***Figure 2—figure supplement 3***, ***Supplementary file 1b***). Out of 20 intact *S. octosporus wtf* candidate genes with confident FIMO hits in intron 1, 17 have detectable short transcripts initiating from within intron 1. Thus, the presence of the FLEX motif in intron 1 correlates with the expression of short transcripts that likely correspond to the *wtf^poison* transcripts (p=0.016, Fisher's exact test). Furthermore, among the 31 genes with detectable short transcripts, those with higher levels of the short transcript are more likely to harbor the FLEX motif in intron 1, as 9 of the top 10 genes ranked by the expression level of the short transcript contain the FLEX motif, whereas only 2 of the bottom 10 genes contain the FLEX motif (p=0.0055, Fisher's exact test). Because 14 *S. octosporus wtf* candidate genes without confident FIMO hits in intron 1 nonetheless do have detectable short transcripts initiating from within intron 1, the lack of a conserved FLEX motif in intron 1 does not appear to preclude the expression of the *wtf^poison* transcript in *S. octosporus*. It is thus possible that some of the candidate *wtf* genes in *S. osmophilus* may also be able to express the *wtf^poison* transcript despite the absence of a high confidence FLEX motif in intron 1.

Interestingly, most of the intact *wtf* candidate genes have an in-frame alternate translational start site near the beginning of exon 2, similar to the *wtf* drivers of *S. pombe* (***Eickbush et al., 2019***; ***Hu et al., 2017***; ***Nuckolls et al., 2017***). The only exceptions are *S. osmophilus wtf16* and S. *cryophilus wtf2*, which appear analogous to the *S. pombe* suppressor *wtf* genes in that they lack an alternate translational start site near the beginning of exon 2 (***Figure 1—figure supplement 1***; ***Bravo Núñez et al., 2018a***; ***Eickbush et al., 2019***). No *wtf* candidate genes appeared similar to the unknown class of *S. pombe wtf* genes (***Supplementary file 1e***; ***Bravo Núñez et al., 2020a***; ***Eickbush et al., 2019***). We note that *wtf* candidate genes in *S. octosporus*, *S. osmophilus,* and *S. cryophilus* share more homology among themselves than they do with *wtf* genes in *S. pombe* (***Supplementary file 1e***).

We conclude, based on amino acid conservation, conserved gene structure, a conserved promoter feature, conserved presence of an alternate transcriptional start site in intron 1, and an alternate translational start site near the beginning of exon 2, that the candidate *wtf* genes we identified in *S. octosporus*, *S. cryophilus,* and *S. osmophilus* are members of the *wtf* gene family. We, therefore, will henceforth refer to them as *wtf* genes.

## *wtf* genes in *S. octosporus, S. osmophilus,* and *S. cryophilus* are associated with dispersed 5S rDNA sequences

The *S. pombe wtf* genes derive their names from their association with solo long terminal repeats (LTRs) of Tf transposons (<u>w</u>ith <u>Tf</u>) (*Bowen et al., 2003*; *Wood et al., 2002*). Most *S. pombe wtf* genes are flanked on at least one side by a solo LTR (*Figure 3*; *Bowen et al., 2003*). A Tf-related full-length transposon was previously discovered in *S. cryophilu*s (designated Tcry1), and we found Tf-related full-length transposons in our *S. osmophilus* assembly (*Rhind et al., 2011*; *Supplementary file 1i*). In *S. cryophilus,* none of the 10 solo LTRs is associated with *wtf* genes. In *S. osmophilus,* 5 out of 36 solo LTRs are associated with *wtf* genes (*Figure 3*). *S. octosporus* does not contain recognizable transposons (*Rhind et al., 2011*).

Instead of a close association with transposon sequences, we found that most of the *wtf* genes outside of *S. pombe* are closely associated with dispersed 5S rDNA genes (*Figure 3*, *Figure 3—figure supplement 1*). In *S. octosporus, S. osmophilus,* and *S. cryophilus,* respectively, 87% (72/83), 79% (33/42), and 40% (2/5) of *wtf* genes are associated with 5S rDNA genes (*Supplementary file 1j*). Conversely, 93% (106/114), 55% (59/107), and 3.4% (4/117) of the 5S rDNA genes in these three species, respectively, are associated with *wtf* genes (*Supplementary file 1j*).

In *S. octosporus, S. osmophilus,* and *S. cryophilus,* we found there is often a gene from an uncharacterized gene family situated between the *wtf* gene and an upstream 5S rDNA gene. We named this new gene family *wag* for <u>w</u>tf-<u>a</u>ssociated <u>g</u>ene (*Figure 3*; *Supplementary file 1b-d and k*). Overall, we found that the genomic context of *wtf* genes could be described by a limited number of patterns, including those first identified in *S. pombe* that are largely specific to that species (*Figure 3*; *Bowen et al., 2003*). These patterns likely reflect a few genomic contexts that were duplicated multiple times during the expansion of the gene family not only as the genes, but also the intergenic sequences within a given type of *wtf*-5S rDNA unit or 5S rDNA-*wag*-*wtf* unit are highly similar within a species (*Figure 3—figure supplements 2–6*).

## *wtf* genes were likely present in the common ancestor of *S. octosporus, S. osmophilus, S. cryophilus,* and *S. pombe*

We next examined whether the *wtf* genes were present in the common ancestor of *S. octosporus, S. osmophilus, S. cryophilus,* and *S. pombe*. The alternate hypothesis is that the *wtf* genes were transferred between the species by horizontal gene transfer or by introgression. Horizontal gene transfer does occur in fission yeast, but the possibility of cross-species introgression is unclear (*Dawe et al., 2018*; *Jeffares et al., 2017*; *Rhind et al., 2011*; *Seike et al., 2019*; *Seike et al., 2015*; *Sipiczki, 1979*; *Sipiczki et al., 1982*).

At the genome level, synteny is limited between *S. pombe* and non-*pombe* fission yeast species (*Rhind et al., 2011*). However, if the *wtf* gene family was vertically inherited from a common ancestor, it is possible that we may find one or more *wtf* loci that exhibit synteny between *S. pombe* and at least one non-*pombe* species. We therefore inspected the genes flanking *S. pombe wtf* genes to look for situations where orthologous genes in another species also flanked a *wtf* gene (*Supplementary file 1l*). We found that in *S. octosporus, S. osmophilus,* and *S. pombe,* one or more *wtf* genes are present between the *clr4* and *met17* genes (*Figure 4A*). This shared synteny could reflect that the ancestor of these species contained a *wtf* gene between *clr4* and *met17,* but it could also mean that the whole *clr4*-*met17* region has undergone horizontal gene transfer or introgression.

To distinguish these possibilities, we analyzed the divergence of the *clr4*-*met17* region between species. Superficially, the region appears quite divergent, with multiple genes gained and/or lost in different lineages. This observation supports a long period of divergence that would be expected if the region descended from the common ancestor of these species. We next analyzed the divergence more precisely. Given the extremely rapid evolution of the *wtf* genes (*Eickbush et al., 2019*), we thought that the flanking genes would prove most informative. If the region was recently transferred between lineages by horizontal gene transfer, it was possible there may be two copies of *clr4* and/or *met17* in the recipient genome. *met17* has an ancient paralog (*SPAC23A1.14c*) present in all fission yeast species, but we found no evidence of recent duplications of *met17* or *clr4*. We also reconstructed phylogenies of the fission yeast *clr4* and *met17* genes and found that the gene trees were consistent with the species trees (*Figure 4B–C*). If the genes had been transferred between species, for example, from the lineage leading to *S. pombe* to the lineage leading to *S. octosporus* and *S. osmophilus,* the

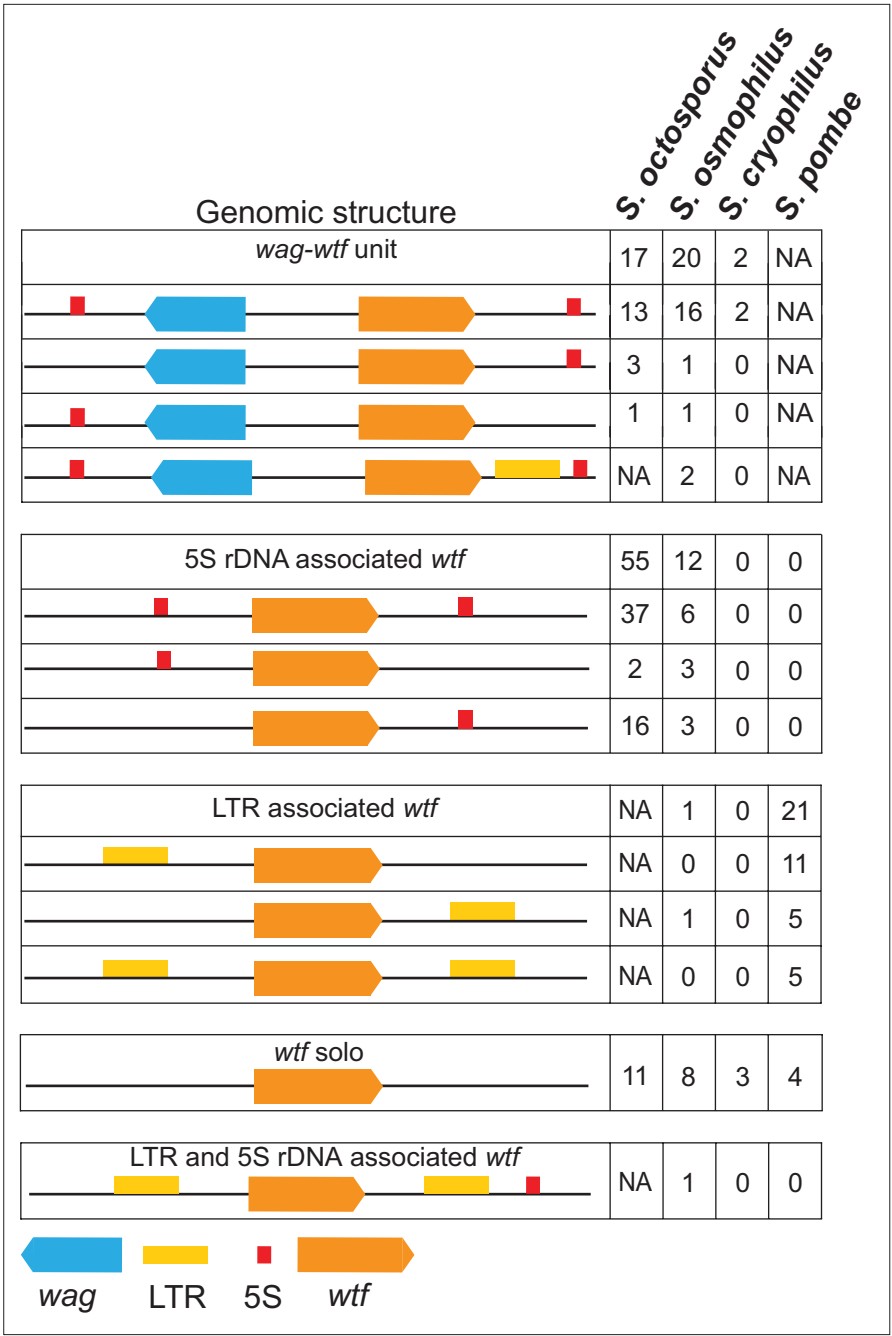

**Figure 3.** Genomic context of *wtf* genes. The *wtf* genes are found in a limited number of genomic contexts. The *wtf* genes are represented as orange boxes, *wag* genes are in blue, and long terminal repeats (LTRs) are in yellow. NA indicates not applicable as *wag* genes are absent from *Schizosaccharomyces pombe,* and LTRs are absent from *S. octosporus.*

The online version of this article includes the following source data and figure supplement(s) for figure 3:

**Figure supplement 1.** Distance between 5S rDNA and *wtf* genes.

**Figure supplement 2.** Homology between distinct 5S rDNA-*wtf* and *wag-wtf* units.

**Figure supplement 2—source data 1.** Multi-alignment of 37 *Schizosaccharomyces octosporus* 5S rDNA-*wtf* units.

**Figure supplement 2—source data 2.** Multi-alignment of 17 *Schizosaccharomyces octosporus wtf-wag* units.

**Figure supplement 3.** *Schizosaccharomyces octosporus wtf* gene units supported by maximum likelihood phylogeny.

*Figure 3 continued on next page*

*Figure 3 continued*

**Figure supplement 4.** Maximum likelihood phylogeny of the regions between *Schizosaccharomyces octosporus wtf* genes and a downstream flanking 5S rDNA gene.

**Figure supplement 4—source data 1.** Multi-alignment of the regions downstream of 67 *Schizosaccharomyces octosporus wtf* with a downstream 5S rDNA.

**Figure supplement 4—source data 2.** Phylogeny of the regions downstream of 67 *Schizosaccharomyces octosporus wtf* genes with a downstream 5S rDNA.

**Figure supplement 5.** Maximum likelihood phylogeny of the regions between *Schizosaccharomyces octosporus wtf* genes and an upstream flanking 5S rDNA gene.

**Figure supplement 5—source data 1.** Multi-alignment of the regions upstream of 40 *Schizosaccharomyces octosporus wtf* with an upstream 5S rDNA.

**Figure supplement 5—source data 2.** Phylogeny of the regions upstream of 40 *Schizosaccharomyces octosporus wtf* genes with an upstream 5S rDNA.

**Figure supplement 6.** Maximum likelihood phylogeny of *Schizosaccharomyces octosporus wtf* genes.

**Figure supplement 6—source data 1.** Multi-alignment of 83 *Schizosaccharomyces octosporus wtf* genes.

**Figure supplement 6—source data 2.** Phylogeny of 83 *Schizosaccharomyces octosporus wtf* genes.

gene tree should reflect that pattern. In this example, the *S. octosporus* and *S. osmophilus clr4* and *met17* genes should group with the *S. pombe* genes on trees, rather than with the *S. cryophilus* genes as we observed. In addition, the percent amino acid divergence we observed in pairwise comparisons between the orthologs revealed divergences similar to the average percent divergences between the species, except for *met17* of *S. octosporus*, which may have gained an intron and diverged extensively (**Supplementary file 1n**). Together, our analyses are consistent with vertical transmission of *clr4* and *met17* and the *wtf* genes between them. This suggests the ancestor of *S. octosporus*, *S. osmophilus*, *S. cryophilus,* and *S. pombe* had a *wtf* gene between *clr4* and *met17* and that the *wtf* gene was lost in the lineage leading to *S. cryophilus* (**Figure 4A**). We found additional shared synteny between *S. pombe wtf6* and *S. cryophilus wtf4*. Again, phylogenetic evidence is consistent with a *wtf* gene being present at that locus in the ancestor of *S. octosporus*, *S. osmophilus*, *S. cryophilus,* and *S. pombe* and being lost in the lineage leading to *S. octosporus* and *S. osmophilus* (**Figure 4—figure supplement 1**, **Supplementary file 1m-n**).

There were additional cases where an *S. pombe wtf* gene was flanked on one side by a gene whose ortholog was also flanked by a *wtf* in one of the other species (**Supplementary file 1l**). We designate this partial synteny. We found three *S. pombe wtf* loci (*wtf27*, the *wtf30+wtf31+wtf10* locus, and *wtf33* all in the *S. kambucha* isolate) with partial synteny with *wtf* genes in *S. octosporus* (*wtf4*, *wtf31*, and *wtf13*; **Supplementary file 1l**; **Eickbush et al., 2019**). Among those three loci, two were also in partial synteny with *wtf* genes in *S. osmophilus* (*wtf5* and *wtf15*). Altogether, our analyses indicate that *S. pombe*, *S. octosporus*, *S. osmophilus,* and *S. cryophilus* inherited *wtf* genes from a common ancestor with multiple *wtf* genes. This does not, however, rule out the possibility that horizontal transfer of *wtf* genes or introgression within fission yeasts also occurred.

## *wtf* genes show evolutionary signatures consistent with a history of genetic conflict

We next wanted to determine if the *wtf* genes are ancient meiotic driver genes or if the genes more recently acquired the ability to drive in the lineage leading to *S. pombe*. To address this, we first analyzed the evolutionary history of the gene family in more depth. Meiotic drivers are predicted to be rapidly evolving, and the *S. pombe wtf* genes support this prediction (**Eickbush et al., 2019**; **Hu et al., 2017**). This rapid evolution is thought to be driven by the genetic conflict predicted to exist between meiotic drivers and the rest of the genome. The conflict arises because the best interest of the driving haplotype (i.e. drive) is at odds with the best interest of the rest of the genome, which is Mendelian allele transmission (**Crow, 1991**). The *wtf* drivers gain an evolutionary advantage by driving, but this is bad for the fitness of the organism because spores are killed (**López Hernández et al., 2021**; **Hu et al., 2017**; **Nuckolls et al., 2017**). The rest of the genome therefore gains an evolutionary advantage by suppressing drive (**Bravo Núñez et al., 2018a**). This conflict is thought to lead

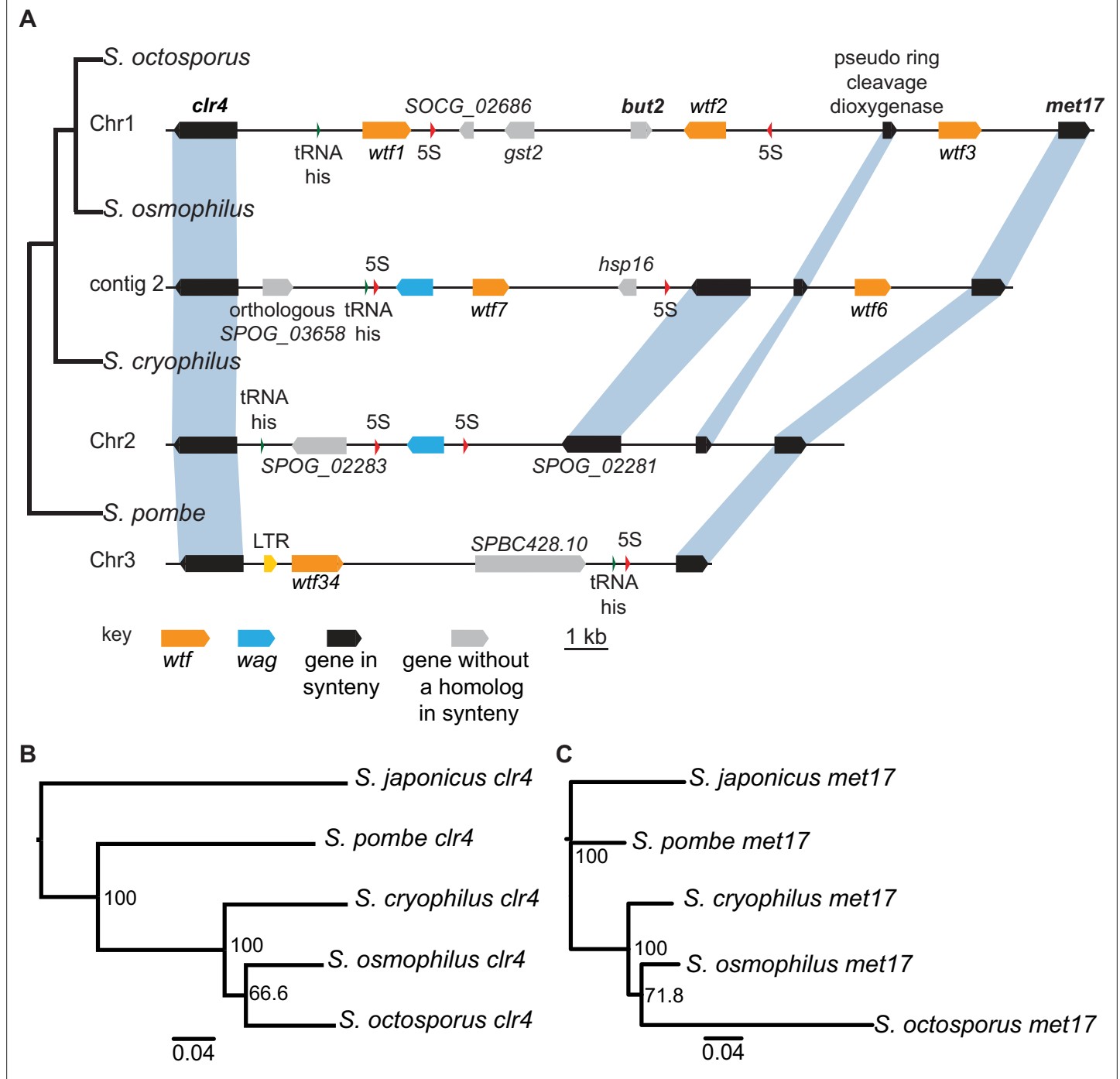

**Figure 4.** Shared *wtf* locus in three fission yeast species. (**A**) The syntenic region between *clr4* and *met17* in *Schizosaccharomyces octosporus*, *S. osmophilus*, *S. cryophilus,* and *S. pombe* is shown. The *S. pombe* locus shown is from the *S. kambucha* isolate. The orange boxes represent *wtf* genes, the blue boxes represent *wag* genes, the red arrows represent 5S rDNA, the green arrow represents *tRNA-his,* the gray boxes represent genes without a homolog in this region in the species shown, and the black boxes represent genes that are syntenic between the species. The phylogenetic relationship between species is shown to the left of the DNA representation. The orthologs of *clr4* (**B**) and *met17* (**C**) were aligned and used to build neighbor-joining trees that were midpoint rooted. Branch support (0–100) was calculated using bootstrap.

The online version of this article includes the following source data and figure supplement(s) for figure 4:

**Source data 1.** Multi-alignment of *Schizosaccharomyces clr4* genes and neighbor-joining tree.

**Source data 2.** Neighbor-joining tree of *Schizosaccharomyces clr4* genes.

**Source data 3.** Multi-alignment of *Schizosaccharomyces met17* genes.

*Figure 4 continued on next page*

*Figure 4 continued*

**Source data 4.** Neighbor-joining tree of *Schizosaccharomyces met17* genes.

**Figure supplement 1.** Synteny between *Schizosaccharomyces cryophilus wtf4* and *S. pombe wtf6*.

**Figure supplement 1—source data 1.** Multi-alignment of *Schizosaccharomyces ago1* genes and neighbor-joining tree.

**Figure supplement 1—source data 2.** Neighbor-joining tree of S*chizosaccharomyces ago1* genes.

**Figure supplement 1—source data 3.** Multi-alignment of *Schizosaccharomyces cyp9* genes.

**Figure supplement 1—source data 4.** Neighbor-joining tree of *Schizosaccharomyces cyp9* genes.

to rapid evolution due to an evolutionary arms race between the drive locus and genomic suppressors where each side must constantly innovate (*McLaughlin and Malik, 2017*).

In *S. pombe*, the evolutionary innovation of *wtf* genes stems from gene duplications, expansion, and contraction of tandem repeats within the coding sequences and extensive non-allelic gene conversion within the family (*Eickbush et al., 2019*; *Hu et al., 2017*). We looked for similar evidence of rapid evolutionary innovation in the *wtf* genes outside of *S. pombe*. As a first step, we built a maximum likelihood phylogeny of intact *wtf* genes from all four species. For *S. pombe*, we used the genes from the FY29033 isolate as it contains more intact *wtf* genes than the reference genome strain. We also excluded genes from the unknown functional class of *S. pombe* (*wtf7*, *wtf11*, *wtf14,* and *wtf15*) because these genes are widely diverged from each other and all other *wtf* genes (*Eickbush et al., 2019*). We observed that the *S. pombe wtf* genes grouped together in a well-supported clade (*Figure 5*, *Figure 5—figure supplement 1*).

For the other three species, the *wtf* genes did not cluster into species-specific monophyletic clades (*Figure 5*, *Figure 5—figure supplement 1*). The *S. cryophilus* genes were found distributed within clades of *S. osmophilus* genes. A total of 37 *S. octosporus* genes grouped together in a well-supported clade. The remaining 11 *S. octosporus* genes grouped together within a well-supported clade that includes 2 *S. osmophilus* genes (*Figure 5*, *Figure 5—figure supplement 1*). Interestingly, this clade of 13 genes contains most (11/14) of the *S. octosporus wag*-associated intact *wtf* genes, and the two *S. osmophilus* genes in the clade are also *wag*-associated. Overall, these patterns are consistent with a history dominated by species-specific duplications and/or species-specific homogenization mediated by non-allelic gene conversion.

We next explored the variation of *wtf* gene numbers to address if the variation is due to extensive overall gene loss since these genes diverged from a common ancestor, variable levels of gene duplication between lineages, or a more complex combination of gene gains and losses. To explore these possibilities, we first returned to our analyses of synteny. If gene loss was the predominant driver of variation in *wtf* gene number, we would expect to find that the *wtf* genes were usually found at a site that also contains a *wtf* gene in one or more additional species. Novel *wtf* gene duplications or horizontal gene transfer events are more likely to be lineage-specific. As described above, all but five *wtf* loci found in *S. pombe* exhibit no synteny in other species (*Supplementary file 1l*). Similarly, there are 31, 12, and 2, species-specific *wtf* loci in *S. octosporus*, *S. osmophilus,* and *S. cryophilus*, respectively (*Supplementary file 1o*). These observations are consistent with novel gene duplications and/ or horizontal gene transfer events occurring in the lineages leading to all four species. Independent expansions are additionally supported by the different genomic contexts of the *wtf* genes in *S. pombe* (Tf-association) and the other species (*wag* and/or 5S rDNA-association). Gene losses are also likely within all lineages, as mentioned above for the loss of ancestral *wtf* gene(s) between *met17* and *clr4* in *S. cryophilus* (*Figure 4A*).

We next looked for signatures of non-allelic gene conversion within the newly discovered *wtf* genes. We started with genes found in synteny with a *wtf* gene in another lineage. These genes should be orthologous and thus group together in a well-supported clade. Non-allelic gene conversion, however, can overwrite genes and thus cause them to be more similar to *wtf* genes at ectopic sites. We focused on *S. octosporus* and *S. osmophilus* as these two species are most closely related, and there are a large number (26) of *wtf* loci showing synteny between these two species (*Supplementary file 1p-q*). We found that none of the genes from syntenic loci group together in a well-supported clade that excludes genes from other loci (*Figure 5*, *Figure 5—source data 1*, *Figure 5—figure supplement 1*). This is consistent with gene conversion frequently overwriting genes in one or both of these two lineages.

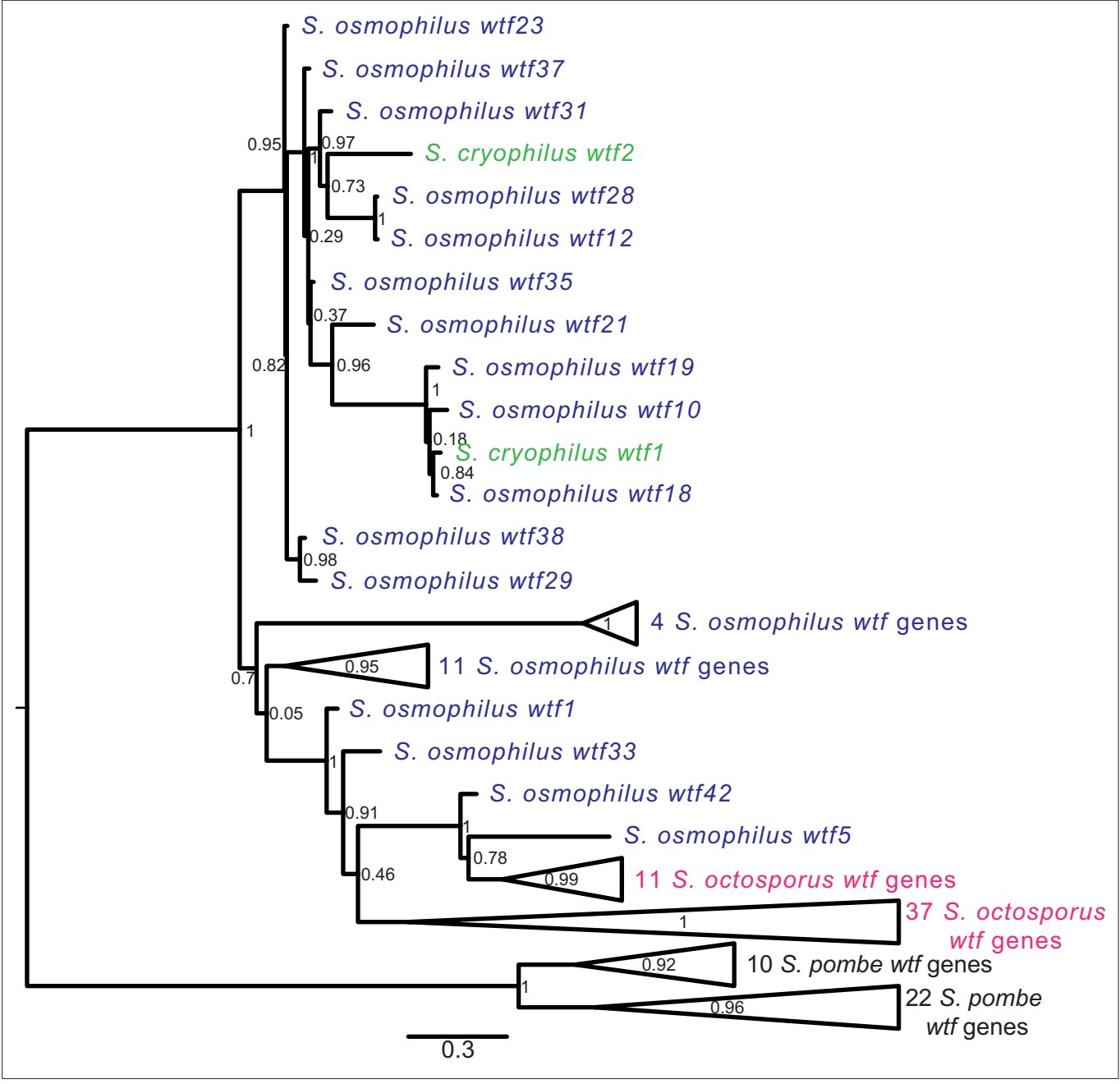

**Figure 5.** Gene duplication and non-allelic gene conversion within *wtf* gene family. All the predicted intact Wtf antidote amino acid sequences were aligned using MAFFT from ***Figure 2—figure supplement 1—source data 1*** and used to build a maximum likelihood tree using PhyML. The *Schizosaccharomyces pombe* sequences were from the FY29033 isolate as it has more *wtf* genes than the reference genome. The *S. pombe* genes are shown in black, *S. octosporus* genes are in magenta, *S. osmophilus* genes are dark blue, and the *S. cryophilus* genes are cyan. The triangles represent multiple genes with the precise number indicated on the right. The branch support values (0–1) are SH-like aLRT values and are shown at each node.

The online version of this article includes the following source data and figure supplement(s) for figure 5:

**Source data 1.** Maximum likelihood phylogeny of 113 *wtf* genes.

**Figure supplement 1.** Gene duplication and non-allelic gene conversion within *wtf* gene family.

**Figure supplement 2.** Genetic algorithm recombination detection (GARD) analysis consistent with non-allelic gene conversion within *wtf* genes.

**Figure supplement 2—source data 1.** Genetic algorithm recombination detection (GARD) analysis of *Schizosaccharomyces octosporus wtf* genes.

**Figure supplement 2—source data 2.** Genetic algorithm recombination detection (GARD) analysis of *Schizosaccharomyces osmophilus wtf* genes.

**Figure supplement 3.** Contraction and expansion of repeat sequences in *wtf* genes.

We next analyzed all the genes within *S. octosporus* and *S. osmophilus* for signatures of gene conversion using the GARD (genetic algorithm recombination detection) program (***Kosakovsky Pond et al., 2006a***). This program builds multiple phylogenetic trees using different segments of genes. If the entire gene shares the same evolutionary history, the trees constructed from different parts of the genes should be the same. Ectopic (non-allelic) gene conversion, however, can shuffle variation within a gene family and lead to differences between trees constructed from different parts of the genes. Consistent with the patterns described above, GARD detected evidence of non-allelic gene conversion within both *S. octosporus* and *S. osmophilus* (***Figure 5—figure supplement 2***).

Finally, we looked for potential evolutionary innovation due to expansion and contraction of tandem repeats within the coding sequences of the newly identified *wtf* genes. Exon 6 of some *S. pombe wtf* genes encodes a 7 amino acid sequence that can be repeated in tandem multiple times (***Eickbush et al., 2019***). An *S. pombe wtf* gene can drive without this sequence, but the number of repeat units found can be important for conferring specificity between a Wtf^poison protein and a Wtf^antidote protein that neutralizes it (***Bravo Núñez et al., 2018a***; ***Nuckolls et al., 2020***). The sequence is thus important for functional innovation of drivers and suppressors. We looked for amino acid repeats in our candidate *wtf* homologs and found a 7 amino acid sequence that was repeated a variable number of times in tandem in exon 4 of genes from *S. octosporus* and *S. osmophilus*. We generated sequence logos to visualize both the nucleotide sequences and amino acid sequences of the repeat in each species (***Figure 5—figure supplement 3***, ***Supplementary file 1r***). We found that the repeat sequences were similar in all three species, consistent with shared ancestry (***Figure 5—figure supplement 3***). For

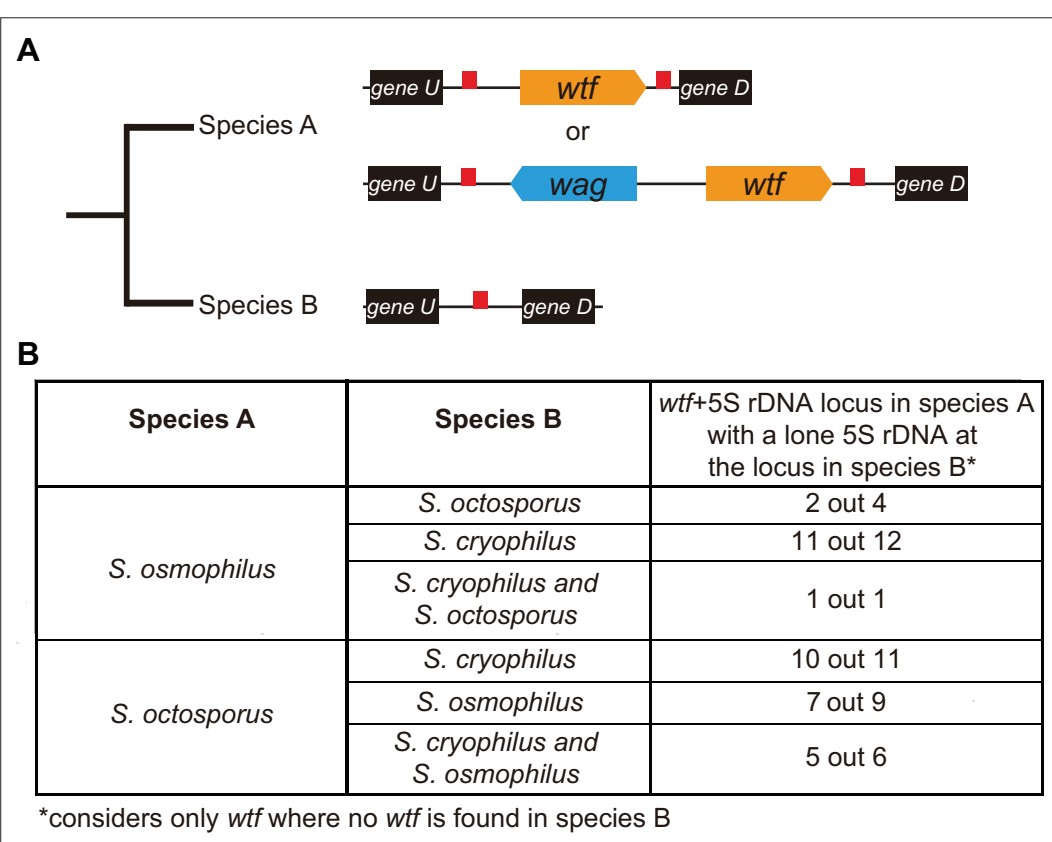

| Species A | Species B | *wtf*+5S rDNA locus in species A with a lone 5S rDNA at the locus in species B* |
|---|---|---|
| *S. osmophilus* | *S. octosporus* | 2 out 4 |
| | *S. cryophilus* | 11 out 12 |
| | *S. cryophilus and S. octosporus* | 1 out 1 |
| *S. octosporus* | *S. cryophilus* | 10 out 11 |
| | *S. osmophilus* | 7 out 9 |
| | *S. cryophilus and S. osmophilus* | 5 out 6 |

*considers only *wtf* where no *wtf* is found in species B

**Figure 6.** *wtf* genes duplicated into pre-existing 5S rDNA. Testing if lineage restricted *wtf* genes occur at sites where the ancestral species is inferred to have had a 5S rDNA gene. An example of this situation is illustrated in (**A**) where species A has a 5S-rDNA-flanked *wtf* gene, and species B has a 5S rDNA gene at the syntenic locus. (**B**) Number of *wtf* + 5S rDNA loci in species A (any of the gene layouts illustrated in (**A**)) with 5S rDNA at the syntenic locus in species B. This analysis only considers loci that contain 5S-rDNA-flanked *wtf* gene in species A but contain no *wtf* genes in species B. Data in ***Supplementary file 1p and q*** were used to test this hypothesis.

The online version of this article includes the following figure supplement(s) for figure 6:

**Figure supplement 1.** *wtf* gene duplication models.

example, the *S. pombe* and *S. osmophilus* repeats both have IGNXXXG as the most common amino acid sequence. The region containing this 7 amino acid repeat exhibits similar length variability in three species (*Figure 5—figure supplement 3*). Hence, like the *S. pombe wtf* drivers, the *wtf* drivers of *S. octosporus* and *S. osmophilus* show signatures of evolutionary innovation via expansion and contraction of a repetitive coding sequence. Together with previous analyses of *S. pombe*, our analyses demonstrate an extensive history of evolutionary innovation within the *wtf* genes. This is consistent with the hypothesis that these genes have a long history as meiotic drivers.

## *wtf* genes duplicated to pre-existing 5S rDNA genes

Given their association with dispersed 5S rDNA genes, we hypothesized that the *wtf* genes in the lineages leading to *S. octosporus*, *S. osmophilus,* and *S. cryophilus* may have duplicated to pre-existing 5S rDNA genes. We propose two recombination models by which this could happen, ectopic gene conversion and integration of extrachromosomal DNA circles (*Figure 6—figure supplement 1A B*; *Cohen et al., 2010*; *Cohen et al., 1999*; *Cohen et al., 2006*; *Cohen and Segal, 2009*; *Daugherty and Zanders, 2019*; *Navrátilová et al., 2008*; *Paulsen et al., 2018*). Under both models, lineage-restricted *wtf* loci flanked with two 5S rDNA genes (e.g. species A in *Figure 6A*) are predicted to have synteny with loci containing a single 5S rDNA gene and no *wtf* genes in other species (e.g. species B in *Figure 6A*). To test this, we first looked at sites where the *S. octosporus* locus contains a 5S-rDNA-flanked *wtf* gene, but the syntenic loci in *S. cryophilus* and *S. osmophilus* do not. There are 6 such *wtf* loci. In 83% (5 out of 6) of those sites, the *S. cryophilus* and *S. osmophilus* loci contain a 5S rDNA gene (*Supplementary file 1q and s*). This is consistent with *wtf* genes duplicating to pre-existing 5S rDNA genes.

We saw similar evidence of *wtf* gene duplication to pre-existing 5S rDNA genes when we considered other species comparisons (*Figure 6B*). For example, we found that in 11 out of 12 sites where *S. osmophilus* has a 5S-rDNA-flanked *wtf* gene but *S. cryophilus* has no *wtf* genes, there is a 5S rDNA gene in *S. cryophilus* (*Figure 6B*, *Supplementary file 1s*). Overall, these analyses support the hypothesis that *wtf* genes spread to pre-existing 5S rDNA genes in the lineages leading to *S. octosporus* and *S. osmophilus.* It is important to note, however, that lineage-specific loss of 5S rDNA-associated *wtf* genes could, and likely does, also contribute to the patterns described above.

## *wtf* genes in *S. octosporus, S. osmophilus,* and *S. cryophilus* encode poison and antidote proteins

We next examined whether there was functional conservation between the *wtf* genes. There are relatively few genetic tools available in fission yeasts outside of *S. pombe*. We therefore first tested the functions of the genes outside of their endogenous species in a more tractable system. We previously demonstrated that the *S. pombe* Wtf4[poison] and Wtf4[antidote] proteins exploit broadly conserved facets of cell physiology and can thus act in the distantly related budding yeast, *Saccharomyces cerevisiae*. Expression of the Wtf4[poison] protein kills vegetative *S. cerevisiae* cells, and co-expression of the Wtf4[antidote] protein neutralizes the toxicity of the Wtf4[poison] protein. Moreover, the mechanisms of the Wtf proteins appear conserved between *S. pombe* and *S. cerevisiae*. Specifically, the Wtf4[antidote] promotes the trafficking of the Wtf4[poison] to the vacuole in both species (*Nuckolls et al., 2020*). We therefore used this established *S. cerevisiae* system to test if the *wtf* genes from the other fission yeast species share functional features with the previously characterized *S. pombe* Wtf4 proteins.

We cloned coding sequences of the putative poison (encoded by exons 2–5) and antidote (encoded by exons 1–5) proteins of *S. octosporus wtf25* and *wtf61*, *S. osmophilus wtf19 and wtf41,* and *S. cryophilus wtf1* under the control of a β-estradiol-inducible promoter on separate plasmids. We then introduced the plasmids into *S. cerevisiae* and analyzed the phenotypes of the resulting strains. We found that induction of each of the putative Wtf[poison] proteins, except *S. osmophilus wtf19*, inhibited cell proliferation in *S. cerevisiae* (*Figure 7A–C*, *Figure 7—figure supplement 1*). Moreover, the toxicity of each functional Wtf[poison] protein was partially neutralized by co-expression of the cognate (i.e. encoded on the same gene) Wtf[antidote] proteins (*Figure 7A–C*; *Figure 7—figure supplement 1B*).

In *S. pombe*, the Wtf[antidote] protein encoded by one *wtf* gene generally cannot neutralize the Wtf[poison] protein encoded by a different *wtf* gene (*Bravo Núñez et al., 2018a*; *Bravo Núñez et al., 2020b*; *Hu et al., 2017*). Instead, a high level of sequence identity appears to be required for a Wtf[antidote] protein to co-assemble with and neutralize a Wtf[poison] protein (*Nuckolls et al., 2020*). We tested if this feature

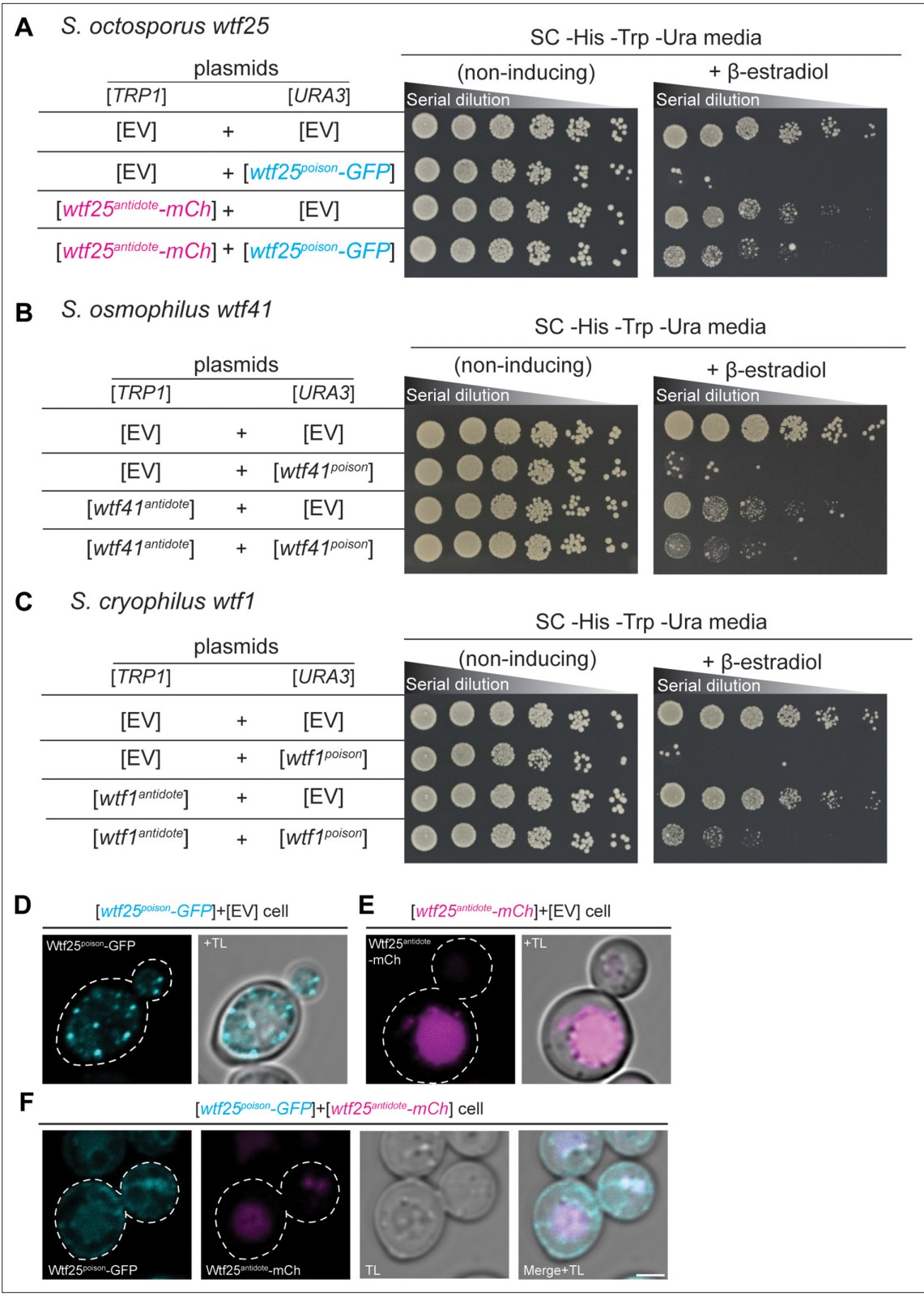

**Figure 7.** *wtf* genes can encode for poison and antidote proteins. Spot assay of serial dilutions of *Saccharomyces cerevisiae* cells on non-inducing (SC -His -Trp -Ura) and inducing (SC -His -Trp -Ura+500 nM β-estradiol) media. Each strain contains [*TRP1*] and [*URA3*] ARS CEN plasmids that are either empty (EV) or carry the indicated β-estradiol inducible *wtf* alleles. (**A**) *Schizosaccharomyces octosporus wtf25poison-GFP* and *wtf25antidote-mCherry* (**B**) *S. osmophilus wtf41poison* and *wtf41antidote*, and (**C**) *S. cryophilus wtf1poison* and *wtf1antidote*. The dilution factor is 0.2 starting at OD = 1. (**D**) A representative

*Figure 7 continued on next page*

*Figure 7 continued*

cell carrying a [*URA3*] plasmid with β-estradiol inducible *S. octosporus wtf25*<sup>poison</sup>-*GFP* (cyan). (**E**) A representative cell carrying a [*TRP1*] plasmid with β-estradiol inducible *S. octosporus wtf25*<sup>antidote</sup>-*mCherry* (magenta). (**F**) A representative *S. cerevisiae* cell carrying a [*URA3*] plasmid with β-estradiol inducible *S. octosporus wtf25*<sup>poison</sup>-*GFP* (cyan) and [*TRP1*] plasmid with β-estradiol inducible *S. octosporus wtf25*<sup>antidote</sup>-*mCherry* (magenta). In all the experiments, the cells were imaged approximately 4 hr after induction with 500 nM β-estradiol. TL = transmitted light. Scale bar represents 2 µm.

The online version of this article includes the following figure supplement(s) for figure 7:

**Figure supplement 1.** Some *wtf* genes outside of *Schizosaccharomyces pombe* encode for poison and antidote proteins.

**Figure supplement 2.** Non-cognate Wtf<sup>antidotes</sup> fail to rescue cells from Wtf<sup>poisons</sup>.

**Figure supplement 3.** The distribution of *Schizosaccharomyces octosporus* Wtf25 proteins is similar to *S. pombe* Wtf4 proteins.

was shared with *wtf* genes outside of *S. pombe*. We tested proteins from five pairs of *wtf* genes. Excluding the antidote protein-specific residues encoded in exon 1, the proteins encoded by each pair share from 13 to 76% amino acid identity. Like our previous observations in *S. pombe*, we found that Wtf<sup>antidote</sup> proteins did not neutralize non-cognate Wtf<sup>poison</sup> proteins (***Figure 7—figure supplement 2A-E***).

To address potential functional conservation of Wtf proteins at higher resolution, we imaged tagged versions of the *S. octosporus* Wtf25 proteins to see if the localization of the proteins in *S. cerevisiae* was similar to what we previously observed for *S. pombe* Wtf4 proteins. *S. octosporus* Wtf25<sup>poison</sup>-GFP and Wtf25<sup>antidote</sup>-mCherry were both functional (***Figure 7A***). *S. octosporus* Wtf25<sup>poison</sup>-GFP was distributed throughout the cytoplasm, with some potential endoplasmic reticulum localization, similar to what we previously observed for *S. pombe* Wtf4<sup>poison</sup>-GFP (***Figure 7D***, ***Figure 7—figure supplement 3***). The *S. octosporus* Wtf25<sup>antidote</sup>-mCherry localization was more restricted. We observed large aggregates outside the vacuole and some signal inside the vacuole (***Figure 7E***, ***Figure 7—figure supplement 3***). This is slightly different from our previous observations with *S. pombe* Wtf4<sup>antidote</sup> as that protein mostly accumulated outside the vacuole in the insoluble protein deposit, with less Wtf4<sup>antidote</sup> protein observed within the vacuole (***Nuckolls et al., 2020***).

When the *S. octosporus* Wtf25<sup>poison</sup>-GFP and Wtf25<sup>antidote</sup>-mCherry proteins were co-expressed, we observed some colocalization of the proteins (***Figure 7F***, ***Figure 7—figure supplement 3***). The colocalized proteins appear to be trafficked to the vacuole. These localization patterns are similar to our previous observations of *S. pombe* Wtf4 proteins where the Wtf4<sup>antidote</sup> co-assembles with the Wtf4<sup>poison</sup> and causes a change of localization of the Wtf4<sup>poison</sup> protein. With *S. pombe* Wtf4 proteins, however, the co-expressed poison and antidote proteins mostly accumulate outside the vacuole at the insoluble protein deposit, with less protein entering the vacuole (***Nuckolls et al., 2020***).

Overall, the poison/antidote functions of the Wtf proteins, the specificity between poison and antidote proteins, and the localization of the Wtf proteins within budding yeast cells we observe in this work are similar to previous observations with *S. pombe* proteins. All together, our data are consistent with broad, but not absolute, functional conservation of the fission yeast Wtf proteins, despite extensive amino acid divergence.

## *wtf* genes can cause meiotic drive in *S. octosporus*

We next formally tested if *wtf* genes could cause meiotic drive outside of *S. pombe* using *S. octosporus*, which among *S. octosporus*, *S. osmophilus,* and *S. cryophilus* is the only one with available genetic tools (***Seike and Niki, 2017***). According to our long-read RNA-seq data, only a small fraction of *wtf* genes in *S. octosporus* have substantial levels of the short transcript isoform (poison isoform) initiated from within intron 1 (***Figure 2—figure supplement 3***). We preferentially tested such genes as we reasoned that a sufficiently high expression level of the poison is essential for drive.

We successfully deleted seven *wtf* genes (*wtf25*, *wtf68*, *wtf33*, *wtf60*, *wtf46*, *wtf21,* and *wtf62*, in the order of decreasing expression levels of the poison isoform) in heterothallic haploid strains of both mating types. No growth phenotypes were observed for these deletion mutants. We then analyzed whether any of the deletions affected viability of spores derived from homozygous and heterozygous crosses using octad dissection analysis (*S. octosporus* generates eight spores per meiosis due to a post-meiotic mitosis prior to spore packaging; ***Chiu, 1996***).

In homozygous crosses, none of the deletions significantly altered spore viability comparing to the wild-type control (***Figure 8***, ***Supplementary file 2a***). Thus, like previous observations for *S. pombe wtf*

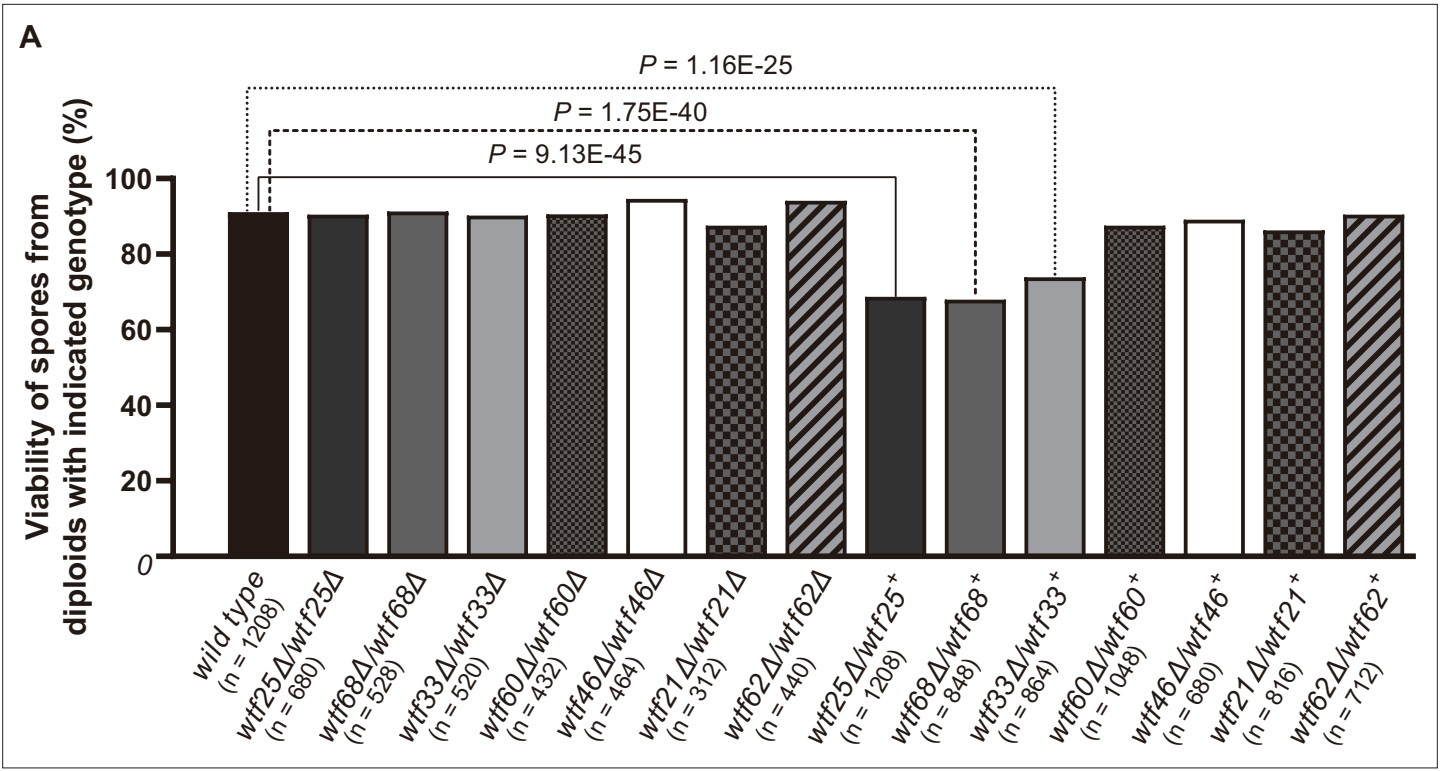

**Figure 8.** Three *Schizosaccharomyces octosporus wtf* genes, when individually deleted, caused spore viability loss in heterozygous crosses but not in homozygous crosses. Deletion mutants of seven *S. octosporus wtf* genes were obtained, and crosses were performed. Heterozygous deletion cross but not homozygous deletion cross of *wtf25*, *wtf68*, or *wtf33* resulted in significant spore viability loss. Spore viability was measured using octad dissection analysis (see Materials and methods). Representative octads are shown in *Figure 9*, *Figure 9—figure supplements 1–6* and *Figure 8* and *Figure 9—source data 2*. Numerical data are provided in *Supplementary file 2b*. p-Values (Fisher's exact test) for crosses with >5% spore viability reduction compared to the wild-type control are shown and calculated in *Figure 8—source data 1*.

The online version of this article includes the following source data for figure 8:

**Source data 1.** Octad analysis tables.

**Source data 2.** Octad dissection raw data.

genes (*Bravo Núñez et al., 2018a*; *Hu et al., 2017*; *Nuckolls et al., 2017*), these seven *S. octosporus wtf* genes are not required for mating, meiosis, or sporulation. In heterozygous crosses, deletion of *wtf25*, *wtf68*, or *wtf33* caused notable and significant spore viability reduction (>5% spore viability reduction and p<0.05, Fisher's exact test). These *wtf+/wtf−* heterozygotes also showed significant drive of the wild-type *wtf+* allele among the viable spores (p<0.05, exact binomial test; *Supplementary file 2b-d*, *Figures 8–9*, *Figure 9—figure supplements 1 and 2*). These results indicate that *wtf25*, *wtf68*, and *wtf33* are active meiotic drivers.

To further explore the octad dissection data, we classified octads derived from heterozygous crosses according to the number of viable spores with a *wtf* gene deletion ('R', antibiotic resistant) and the number of viable spores without a *wtf* gene deletion ('S', antibiotic sensitive) in an octad. For example, an octad with seven viable spores can be classified as either the 4R3S type or the 3R4S type. If spore viability is not affected by *wtf* gene deletion, the ratios of 4R3S to 3R4S, 4R2S to 2R4S, 4R1S to 1R4S, and 4R0S to 0R4S should be about 1:1. For *wtf25*, *wtf68*, and *wtf33*, the three genes deemed as active meiotic drivers based on the analysis of overall spore data, most of these octad-type ratios significantly deviate from 1:1 (p<0.05, exact binomial test; *Figure 9C*, *Figure 9—figure supplements 1 and 2*). The 4R2S to 2R4S ratio of *wtf60* and the 4R3S to 3R4S ratio of *wtf46* also significantly deviate from 1:1 (*Figure 9—figure supplements 3 and 4*, *Supplementary file 2e-f*), suggesting that *wtf60* and *wtf46* may have weak meiotic driver activities. *wtf21* and *wtf62* did not cause significant deviation of octad-type ratios (*Figure 9—figure supplements 5 and 6*, *Supplementary file 2g-h*), consistent with the low expression levels of the poison isoforms of these two genes (*Figure 2—figure*

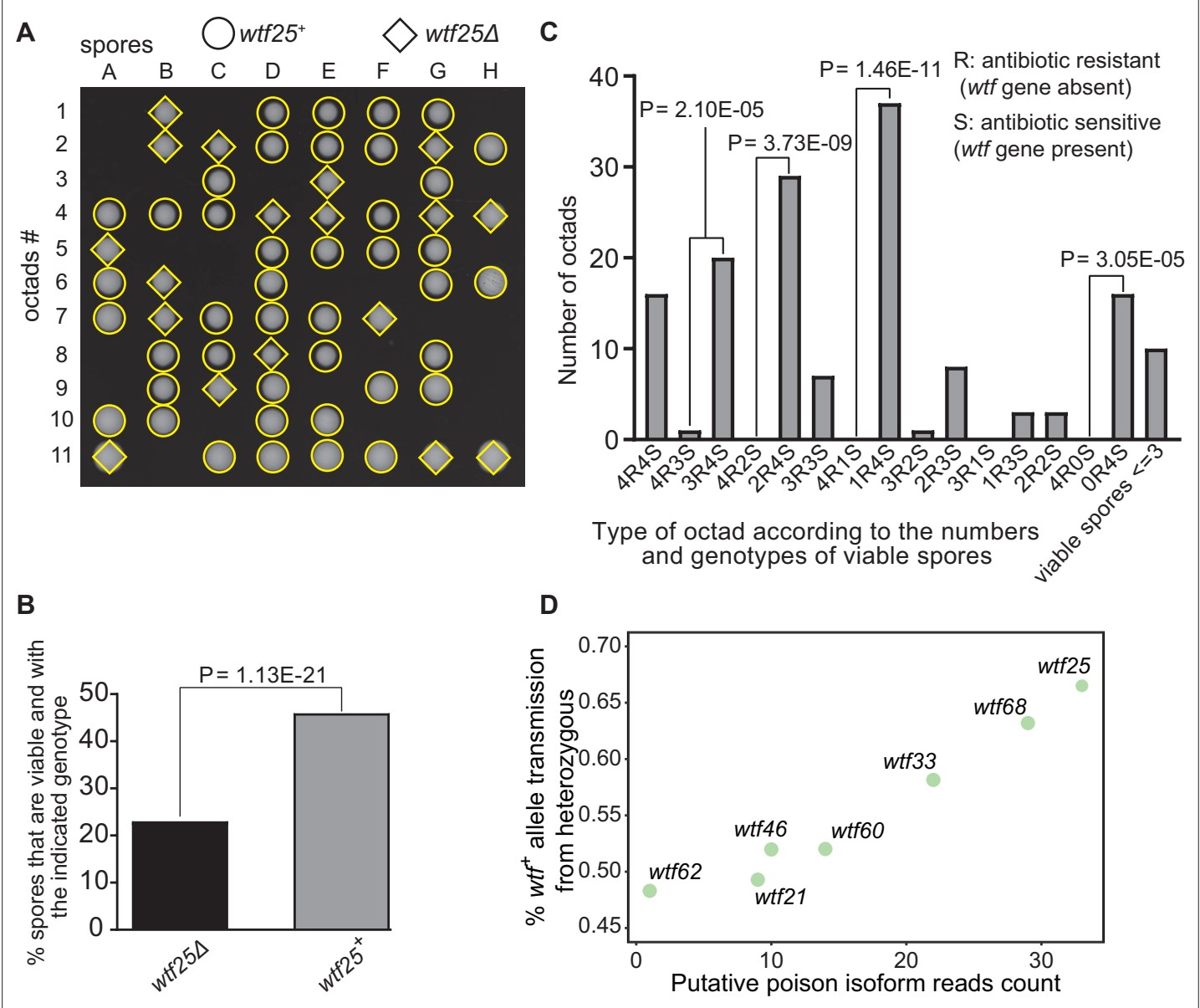

**Figure 9.** Some *Schizosaccharomyces octosporus wtf* genes cause meiotic drive. (**A**) Representative octads dissected from asci produced from a *wtf25* heterozygous deletion cross. The labels A–H indicate the 8 spores dissected from each ascus, and the labels 1–11 indicate the 11 asci analyzed. The genotypes of clones were determined by replica plating onto antibiotic-containing plates. Raw data of all octads can be found in *Figure 9—source data 2*. (**B**) The percentages of spores that were viable and with indicated genotypes produced by *wtf25⁺*/*wtf25Δ* cross. The p-value was calculated using exact binomial test, and numerical data are provided in *Figure 9—source data 1*. (**C**) Classification of octads derived from *wtf25⁺*/*wtf25Δ* cross according to the number of viable spores with and without a *wtf* gene deletion. The p-values were calculated using the exact binomial test. The p-values are only displayed if a pair of octad types have more than five octads in total, as p-values cannot reach the significance threshold if the total number of octads ≤5. (**D**) Correlation between transmission distortion ratio and poison isoform expression level. The transmission distortion ratio represents the proportion of *wtf* containing spores in total viable spores produced by a *wtf⁺*/*wtfΔ* heterozygote, and the read counts are those shown in *Figure 2—figure supplement 3*. Numerical data of transmission distortion ratio of each *wtf* gene can be found in *Supplementary files 2b-h*.

The online version of this article includes the following source data and figure supplement(s) for figure 9:

**Source data 1.** Numerical data of the octad dissection analysis of *wtf25* heterozygous deletion cross.

**Source data 2.** *wtf25* heterozygous diploid octad dissection raw data.

**Source data 3.** *wtf25* homozygous diploid octad dissection raw data.

**Figure supplement 1.** Octad dissection analysis of *wtf68* heterozygous deletion cross.

*Figure 9 continued on next page*

*Figure 9 continued*

**Figure supplement 1—source data 1.** Numerical data of the octad dissection analysis of *wtf68* heterozygous deletion cross.

**Figure supplement 1—source data 2.** *wtf68* heterozygous diploid octad dissection raw data.

**Figure supplement 1—source data 3.** *wtf68* homozygous diploid octad dissection raw data.

**Figure supplement 2.** Octad dissection analysis of *wtf33* heterozygous deletion cross.

**Figure supplement 2—source data 1.** Numerical data of the octad dissection analysis of *wtf33* heterozygous deletion cross.

**Figure supplement 2—source data 2.** *wtf33* heterozygous diploid octad dissection raw data.

**Figure supplement 2—source data 3.** *wtf33* homozygous diploid octad dissection raw data.

**Figure supplement 3.** Octad dissection analysis of *wtf60* heterozygous deletion cross.

**Figure supplement 3—source data 1.** Numerical data of the octad dissection analysis of *wtf60* heterozygous deletion cross.

**Figure supplement 3—source data 2.** *wtf60* heterozygous diploid octad dissection raw data.

**Figure supplement 3—source data 3.** *wtf60* homozygous diploid octad dissection raw data.

**Figure supplement 4.** Octad dissection analysis of *wtf46* heterozygous deletion cross.

**Figure supplement 4—source data 1.** Numerical data of the octad dissection analysis of *wtf46* heterozygous deletion cross.

**Figure supplement 4—source data 2.** *wtf46* heterozygous diploid octad dissection raw data.

**Figure supplement 4—source data 3.** *wtf46* homozygous diploid octad dissection raw data.

**Figure supplement 5.** Octad dissection analysis of *wtf21* heterozygous deletion cross.

**Figure supplement 5—source data 1.** Numerical data of the octad dissection analysis of *wtf21* heterozygous deletion cross.

**Figure supplement 5—source data 2.** *wtf21* heterozygous diploid octad dissection raw data.

**Figure supplement 5—source data 3.** *wtf21* homozygous diploid octad dissection raw data.

**Figure supplement 6.** Octad dissection analysis of *wtf62* heterozygous deletion cross.

**Figure supplement 6—source data 1.** Numerical data of the octad dissection analysis of *wtf62* heterozygous deletion cross.

**Figure supplement 6—source data 2.** *wtf62* heterozygous diploid octad dissection raw data.

**Figure supplement 6—source data 3.** *wtf62* homozygous diploid octad dissection raw data.

*supplement 3*). In fact, the levels of allele transmission bias favoring the *wtf+* allele appear to be correlated with the expression levels of the poison isoform (*Figure 9D*).

## *S. octosporus wtf25* is a poison and antidote meiotic driver

To determine whether an active *wtf* gene in *S. octosporus* can cause meiotic drive at an ectopic genomic locus, we constructed an integrating plasmid carrying a 2.5-kb genomic region containing *wtf25* together with its upstream and downstream flanking 5S rDNA genes and integrated the plasmid at the *leu1* locus in the *wtf25* deletion background. Octad dissection analysis indicated that *wtf25* integrated at the *leu1* locus can act as a meiotic driver in a heterozygous cross (*leu1Δ::wtf25/leu1*), and the level of meiotic drive was comparable to that caused by the endogenous *wtf25* gene (*Figure 10B*). This result indicates that *S. octosporus wtf25* can act in a locus-independent manner like the *S. pombe wtf* drivers. *wtf25* can express a long transcript isoform and a short transcript isoform through alternative transcriptional initiation (*Figure 2B*). Based on what is known about the *S. pombe wtf* genes and our analyses of *S. octosporus wtf25* in *S. cerevisiae* (*Figure 7*), we hypothesized that the long and short isoforms encode antidote and poison proteins, respectively (*Hu et al., 2017*; *Nuckolls et al., 2017*). We introduced point mutations into the predicted start codons of the long and short isoforms of *wtf25* integrated at the *leu1* locus and analyzed the effects of the mutations on spore viability (*Figure 10A*). To disrupt the predicted Wtf25[poison] protein coding sequence, we mutated the predicted start codon (ATG to GCG, methionine to alanine) in the short transcript to generate the *wtf25[antidote-only]* allele. As expected, this allele was unable to kill spores not inheriting it in a *wtf25* deletion background (*Figure 10B*, *Supplementary file 2i*). This supports our hypothesis that the short transcript encodes a spore-killing poison.

To disrupt the predicted Wtf25[antidote] protein coding sequence, we mutated the predicted start codon in the long transcript isoform to generate the *wtf25[poison-only]* mutant allele (*Figure 10A*). We could not obtain transformants of the plasmid carrying this mutant allele in the *wtf25* deletion background, possibly due to self-killing. Instead, we integrated the plasmid at the *leu1* locus in the

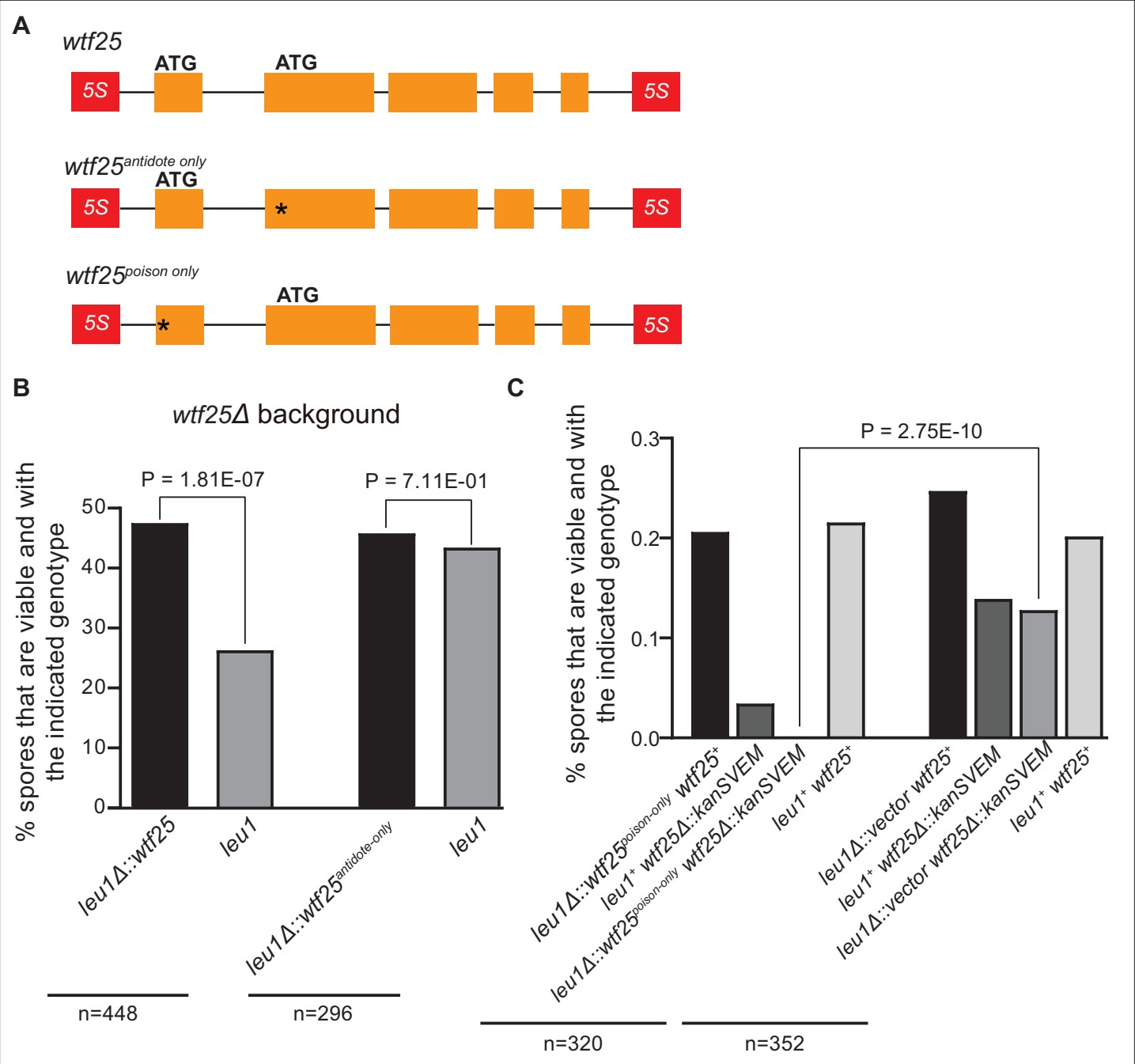

**Figure 10.** *Schizosaccharomyces octosporus wtf25* is a poison-and-antidote killer meiotic driver. (**A**) Schematic of the *wtf25* alleles integrated at the *leu1* (*SOCG_02003*) locus. Black asterisks indicate start codon mutations. The start codon for the putative *wtf25poison* coding sequence is mutated in the *wtf25antidote-only* allele, and the start codon for the putative *wtf25antidote* coding sequence is mutated in the *wtf25poison-only* allele. (**B**) The wild-type *wtf25* allele integrated at the *leu1* locus can act as a meiotic driver by killing spores not inheriting it in a heterozygous cross, while *wtf25antidote-only* mutant allele integrated at the same locus was unable to kill spores not inheriting it in a heterozygous cross. p-Value calculations using a binomial test of goodness-of-fit are shown in *Figure 10—source data 1 and 2*. (**C**) The *wtf25poison-only* allele integrated at *leu1* can cause self-killing in spores that do not inherit wild-type *wtf25* at the endogenous locus. The effects of the *wtf25poison-only* allele were compared to a control cross in which an empty vector was integrated at *leu1*. Numerical data are provided in *Supplementary file 2i*, and the p-value calculation is shown in *Figure 10—source data 3*.

The online version of this article includes the following source data for figure 10:

**Source data 1.** Raw data of the octad dissection analysis of *wtf25* integrated at *leu1*.

**Source data 2.** Raw data of the octad dissection analysis of *wtf25antidote-only* integrated at *leu1*.

**Source data 3.** Raw data of the octad dissection analysis of *wtf25poison-only* integrated at *leu1*.

wild-type background and crossed the resulting strain to a *wtf25Δ* strain. As a control, we integrated an empty vector at the *leu1* locus in the wild-type background and crossed the resulting strain to a *wtf25Δ* strain. Compared to the control, *wtf25Δ* spores (spores not inheriting the wild-type *wtf25* at the endogenous locus) derived from diploids carrying the *wtf25^poison-only* allele suffered markedly more severe viability loss (**Figure 10C**). Among them, the *wtf25Δ* spores that also inherited the *wtf25^poison-only* mutant allele at the *leu1* locus were all inviable. These results further support the model that the short isoform encodes a poison protein that confers killing but not protection. In addition, they demonstrate that the long isoform is required for protection against spore killing.

## Discussion

### *wtf* genes are ancient meiotic drivers

Our analyses indicate that *wtf* genes were present in the common ancestor of *S. octosporus*, *S. osmophilus*, *S. cryophilus,* and *S. pombe*. As these species are estimated to have diverged ~119 million years ago (**Rhind et al., 2011**), we propose that the *wtf* gene family is over 100 million years old. Our results suggest that ancestral *wtf* genes could cause meiotic drive, and we propose that *wtf* genes have at least occasionally caused meiotic drive throughout their history. First, the gene family exhibits several signatures of the rapid evolutionary innovation typified by genes involved in genetic conflicts (**Figure 5**, **Figure 5—figure supplements 1–3**; **Burt and Trivers, 2006**; **McLaughlin and Malik, 2017**). Also, genes from all four species encode both poison and antidote proteins, like the known drivers in *S. pombe* (**Figure 7**). In addition, genes from at least three species contain the FLEX regulatory motif upstream of the open reading frame that encodes a poison protein, suggesting the genes are expressed in meiosis (**Figure 2**, **Figure 2—figure supplement 2**). Our RNA sequencing data confirms this hypothesis in *S. octosporus* (**Figure 2B**, **Figure 2—figure supplement 3**). And finally, we demonstrate that some *S. octosporus wtf* genes cause meiotic drive when heterozygous (**Figure 9**).

We have been unable to trace the history of the *wtf* gene family farther back than the ancestor of *S. octosporus*, *S. osmophilus*, *S. cryophilus,* and *S. pombe*. It is possible that the genes were born de novo within this lineage. Alternately, it is possible the genes may also have entered the lineage via horizontal gene transfer. Distinguishing these possibilities will likely prove difficult. The old age and rapid evolution of the gene family largely restrict our ability to reconstruct the sequence of the ancestral gene(s) with confidence. In addition, given that the genes encode poison and antidote proteins, it is possible that any potential extant homologs outside of fission yeast will have experienced a history of genetic conflict and could be quite diverged from their ancestral state as well.

### Repeat facilitated expansion of the *wtf* gene family

Our synteny analyses indicate that the ancestor of the four species contained at least 2 *wtf* genes (**Figure 4**, **Figure 4—figure supplement 1**, **Supplementary file 1I**), and the extant species carry between 5 and 83 *wtf* genes, including pseudogenes (**Figure 1B**). Our analyses are consistent with novel gene duplications occurring in the lineages leading to all four species. The *wtf* genes are compact and can autonomously cause drive. These features likely facilitated their spread within genomes. In this study, we show that non-allelic recombination using repetitive 5S rDNA sequences has likely facilitated the expansion of the *wtf* gene family in *S. octosporus* and *S. osmophilus*. This recombination could be non-allelic gene conversion but could also be crossovers involving extrachromosomal circles as many *wtf* genes are flanked by direct repeats of 5S rDNA genes (**Figure 3**, **Figure 6**). The later pathway was recently implicated in the spread of Rsp-like sequences in *Drosophila* species (**Sproul et al., 2020**). The newly formed *wtf* gene duplicates could be maintained at a high rate due to their potential to cause drive or to suppress drive of other *wtf* genes with a similar sequence.

It may be relevant that both the Tf LTRs and the dispersed 5S rDNA genes cluster spatially in the nucleus (**Cam et al., 2008**; **Iwasaki et al., 2010**). The Tf LTR transposons are bound by CENP-B family proteins and are clustered to a nuclear domain known as the Tf body in a process that requires the CENP-B family protein Cbp1, the histone lysine H3-K4 methyltransferase Set1, and the Ku protein Pku80 (**Johansen and Cam, 2015**). Interestingly, 5S rDNA genes are also bound by Cbp1 and can cluster with other pol III transcribed genes within the nucleus (**Daulny et al., 2016**; **Iwasaki et al., 2010**). Such clustering may promote the duplication of *wtf* genes to novel repeat-associated sites in the genome due to physical proximity. Speculatively, the clusters could also potentially affect

recombination outcomes. For example, proteins found in the clusters could promote DNA repair pathways that lead to gene conversion rather than crossovers. In this way, the clusters could facilitate non-allelic gene conversion that helps enable the rapid evolution of the *wtf* genes (*Eickbush et al., 2019*; *Hu et al., 2017*), while limiting non-allelic crossovers that generate costly chromosome rearrangements.

It is also interesting to note that *Rsp-like* sequences of *Drosophila* mentioned above also duplicated to distributed repetitive sequences (1.688 satellite) that Hi-C data suggest may cluster within nuclei (*Sproul et al., 2020*). Furthermore, genes of the *Dox* gene family in *D. simulans* are associated with 359-base pair satellite, a member of the 1.688 satellite family, which has been proposed to have facilitated expansion of the family (*Muirhead and Presgraves, 2021*; *Vedanayagam et al., 2021*). Experimental analyses of the effect of clustered repeats on sequence duplication and ectopic recombination outcomes are required to explore how these sequences affect recombination (*Muirhead and Presgraves, 2021*; *Vedanayagam et al., 2021*).

## Potential factors contributing to long-term persistence of *wtf* drivers

The *wtf* drivers are part of a large, rapidly evolving gene family that also includes *wtf* genes that act as drive suppressors (*Bravo Núñez et al., 2018a*). We propose that this multi-gene landscape might enable a cycle of driver death and rebirth analogous to the mythological phoenix. *wtf* drivers could be perpetually reborn anew via gene duplication and rapid evolution of existing genes. This rebirth could allow the gene family to evade extinction by disrupting or reversing the two main paths to extinction mentioned earlier: extinction following suppression and extinction following fixation. In this way, drive is not tied to a single gene or locus over long evolutionary timescales. In fact, all four genomes contain *wtf* pseudogenes that could be degraded versions of past drivers. Instead, the genetic capacity for drive is a property that is maintained by the gene family.

Understanding the rebirth cycle we propose requires mechanistic understanding of Wtf proteins. Specifically, in all known cases, a given Wtf[antidote] protein only neutralizes Wtf[poison] proteins with amino acid sequences that are highly similar to that of the Wtf[antidote] (*Hu et al., 2017*; *Bravo Núñez et al., 2020b*). This is likely because homotypic assembly underlies the Wtf[poison]/Wtf[antidote] co-assembly that promotes neutralization of the poison by trafficking it to the vacuole (*Nuckolls et al., 2020*). Likely using this mechanism, *wtf* drivers can be suppressed by other *wtf* genes encoding an antidote like that of the driver. The *wtf* suppressors can be genes that only encode antidotes are thus non-autonomous drivers (*Bravo Núñez et al., 2018a*), but complete *wtf* drivers can also suppress each other (*Bravo Núñez et al., 2020a*). Importantly, mutations in a *wtf* driver that generate a novel poison simultaneously generate a novel compatible antidote, as the two proteins are encoded on overlapping coding sequences.

The mechanism of Wtf proteins described above means that mutations in *wtf* genes can often generate new proteins with novel assembly properties. Rather than being doomed to extinction, a *wtf* driver that is fixed or suppressed within a local population could be reborn via mutation. For example, changes in intragenic repeat copy number are predicted to generate novel drivers and suppressors (*Bravo Núñez et al., 2018a*, *Nuckolls et al., 2020*). The many copies of *wtf* genes, however, could be even larger contributors to this proposed rebirth process due to intrafamily gene conversion. Such gene conversion could even lead to rebirth of an active driver or suppressor at a locus previously encoding a *wtf* pseudogene.

It is important to note that our proposed model for evolutionary rebirth of *wtf* gene has several critical limitations. First, the timescale of the process is unclear. Modeling and experimental evolution analyses have demonstrated that a novel *wtf* driver can quickly spread to fixation in a local population (*López Hernández et al., 2021*). However, the ecology of *S. pombe* is not well understood (*Jeffares, 2018*), including how often cells encounter outcrossing partners that are not closely related by clonal decent. Such encounters would be required for a driver to spread globally to species-level fixation. An additional limitation is that we do not know the rate at which *wtf* genes acquire mutations that lead to novel drive phenotypes. Still, analyses of the *wtf* gene landscape in *S. pombe* isolates suggest that the mutation rate of the gene family is high. Despite *S. pombe* genomes sharing >99% DNA sequence identity (*Jeffares et al., 2015*), the *wtf* genes found in the four assembled isolates are generally distinct (*Hu et al., 2017*; *Eickbush et al., 2019*). In fact, there is only one locus, *wtf4*, where all four isolates contain a driver (*Eickbush et al., 2019*). The sequence of the *wtf4* driver, however, is not fixed

globally in the species. Even the two most similar *wtf4* drivers, from the reference genome (*Sp*) and the *S. kambucha* isolate (*Sk*), are distinct drivers in that the antidote from *Sp wtf4* does not neutralize the poison from *Sk wtf4* and vice versa (**Nuckolls et al., 2017**). Thus, *wtf* gene sequences may change faster than they can spread globally to fixation in a species.

An alternate hypothesis to explain the long-term persistence of the *wtf* drivers is that the genes are not merely selfish parasites. Importantly, analyses of strains in which all *wtf* genes have been deleted have not yet been reported. It is possible that *wtf* genes promote fitness in some way that analyses of such strains could reveal. The potential beneficial function(s) of the *wtf* genes could have promoted their long-term maintenance in fission yeast genomes. It is important to note, however, that genes do not need to promote fitness to be maintained in genomes.

## Materials and methods

### Nanopore sequencing and assembly of the *S. osmophilus* genome

To sequence the genome of *S. osmophilus* strain CBS 15792, we extracted genomic DNA with the QIAGEN genomic-tip kit. We then used a standard ligation sequencing prep and kit (SQK-LSK109; Oxford Nanopore Technologies [ONT]), including DNA end repair using the NEB End Prep Enzyme, FFPE prep with the NEB FFPE DNA repair mix, and ligation using NEB Quick Ligase. We sequenced using two Flongle sequencers and performed base calling with guppy version 2.1.3. This generated approximately 521 megabases of sequence or approximately 40× coverage. We then performed de novo assembly pathway using canu v1.8 and the ovl overlapper with a predicted genome size of 13 mb and a corrected error rate of 0.12. We corrected our assembly using pilon with paired-end illumina data generated with the same DNA. We assembled 11 nuclear contigs with a total length of 11.3 mb and one mitochondrial contig that was 68 kb in length. Assembly statistics were generated using an existing perl script (https://github.com/SchwarzEM/ems_perl/blob/master/fasta/count_fasta_residues.pl, **Lee et al., 2021**). The assembled genome scored at 89% complete with BUSCO v3.0.2, which is comparable to the score for the closely related species *S. octosporus* (**Simão et al., 2015**). Base called reads were deposited on the SRA under project accession code PRJNA839783.

### RNA sequencing and nanopore cDNA sequencing

Sample preparation

For RNA sequencing and ONT cDNA sequencing of *S. octosporus* diploid cells undergoing azygotic meiosis, we crossed strains DY44617 and DY44598 (**Supplementary file 3a**) on a SPASK plate (1% glucose, 7.3 mM $KH_2PO_4$, vitamins, 45 mg/L of leucine, adenine, uracil, histidine, and lysine) for about 12 hr. Cells were spread on YES plates (0.5% yeast extract, 3% glucose, 200 mg/L of leucine, adenine, uracil, and histidine) containing nourseothricin (NAT) and G418 (YES+NAT&G418) for diploid cell selection. After 3 days, colonies grown up on YES+NAT&G418 plates were collected and spread on YES plates. After 24 hr, cells were washed off from YES plates and spread on SPASK plates for azygotic meiosis induction. Approximately 5 $OD_{600}$ units of cells were harvested and snap frozen using liquid nitrogen 19 hr after the start of meiosis induction.

RNA extraction

All collected cells were thawed on ice for about 5 min and then washed once with chilled DEPC water. The cell pellets were resuspended with TES buffer (10 mM Tris pH 7.5, 10 mM EDTA pH 8.0, 0.5% SDS) and mixed with acidic phenol-chloroform (1:1) immediately. The samples were incubated in a 65°C heat block for 1 hr. Then the samples were centrifuged at 4°C, and the aqueous phase was collected. The aqueous phase was then treated with phenol-chloroform (1:1) and chloroform:isoamyl alcohol (24:1) successively. 3 M NaAc (pH 5.2) and isopropanol were added to the aqueous phase and mixed thoroughly by inverting. The mixture was stored at –20°C overnight and then centrifuged at 4°C. After centrifuging, the supernatants were removed, and the RNA pellets were washed with 70% ethanol. RNA samples were dissolved in DEPC water after air-drying.

RNA sequencing

For RNA sequencing, we prepared total RNA from *S. octosporus* cells undergoing azygotic meiosis as described above. Sequencing library construction and Illumina 150 bp paired-end sequencing were

performed by Annoroad Gene Technology (Beijing, China). The raw sequencing reads were processed using fastp (version:0.20.0), with default parameters. The cleaned reads were mapped to a high-quality *S. octosporus* reference genome (http://bifx-core.bio.ed.ac.uk/~ptong/genome_assembly/oct_genome.fa) using STAR (version: 2.6.0 a) with the following settings: '--alignIntronMin 29 --alignIntronMax 819 --outFilterMultimapNmax 1 --outFilterMismatchNmax 0 --alignEndsType EndToEnd' (*Dobin and Gingeras, 2016*). The strains used to prepare the RNA are derived from the reference genome strain and are thus highly similar. Illumina sequencing data were deposited at NCBI SRA under the accession number SRR17543073.

## Long-read cDNA sequencing

For long-read cDNA sequencing using the ONT platform, we prepared total RNA as described above. Sequencing library construction and ONT cDNA sequencing were performed by Biomarker Technologies (Qingdao, China). Through processing the reads using pychopper (version 2.3.1), we obtained 2,839,411 full-length reads. We performed further data analysis using FLAIR (full-length alternative isoform analysis of RNA, version 1.5) (*Tang et al., 2020*). FLAIR was designed to perform reads mapping, reads correcting, and isoform clustering for noisy long reads generated by ONT cDNA sequencing, and it can be run optionally with short-read RNA sequencing data to help increasing the accuracy of splicing site identification in isoforms. We mapped full-length reads to the *S. octosporus* reference genome mentioned above using 'flair.py align' submodule with default parameters. The splicing junction information generated by short-read RNA sequencing was first extracted using a FLAIR script called 'junctions_from_sam.py' from the reads mapping SAM file then submitted to 'flair.py correct' submodule. Finally, we generated high-quality transcript information by running 'flair.py collapse' submodule with default parameters (*Tang et al., 2020*). The ONT cDNA reads mapping results were visualized using the Integrative Genomics Viewer (*Robinson et al., 2011*; an example is shown in *Figure 2B*), and the transcripts generated by FLAIR were used for *wtf* and *wag* gene structure annotation polishing. ONT cDNA sequencing data were deposited at NCBI SRA under the accession number SRR17543072.

## *S. osmophilus* genome annotation

For *S. osmophilus,* we annotated all the coding sequences with the Augustus gene prediction software webpage (*Stanke et al., 2006*; http://bioinf.uni-greifswald.de/webaugustus/). First, we trained Augustus software with *S. octosporus* genome from *Tong et al., 2019*, and we uploaded the cDNA sequences of *S. octosporus* genes from *Rhind et al., 2011*. This training set allowed Augustus to construct a model then to predict *S. osmophilus* genes. Augustus annotated the predicted exons and introns of all the genes in *S. osmophilus* genome found in *Figure 1—source data 2*. To match *S. osmophilus* genes with orthologous genes within *S. octosporus*, *S. cryophilus*, and *S. pombe*, we extracted all the predict translations of *S. osmophilus* genes and used OrthoVenn2 to find orthologs for each gene (*Xu et al., 2019*). The orthologs are reported in *Figure 1—source data 3*.

## Calculating amino acid identity between *Schizosaccharomyces* species

To calculate the percentage amino acid identity shared between proteins of the different *Schizosaccharomyces* species, we used BLASTp (default parameters) to compare each protein sequence to a protein database created for each genome (*Altschul et al., 1990*). For example, we compared all the genes of *S. osmophilus* with the *S. octosporus* database. We then compared all the genes of *S. octosporus* with the *S. osmophilus* database. The best hit was saved for each gene from the reciprocal BLASTp to calculate the percentage of identity between two orthologs (*Figure 1—source data 3*). We then calculate the mean of all the percentage identity (all the genes) between the two genomes. The percentage of identity for each paired comparisons between genomes can be found in *Supplementary file 1a*. This percentage of identity was used to verify the previously proposed *Schizosaccharomyces* phylogeny, based on limited data from *S. osmophilus*, shown in *Figure 1B*; *Brysch-Herzberg et al., 2019*; *Rhind et al., 2011*.

## Sequence homology search

To find *wtf* genes outside of *S. pombe,* we performed a PSI-BLAST search within the *Schizosaccharomyces* species with the *S. pombe wtf4* gene as a query (E-value threshold 0.05, word size: 3, matrix:

BLOSUM62, gap existence: 11, gap extension: 1, PSI-BLAST threshold: 0.005; *Altschul et al., 1997*). We repeated the search until no new significant hits were found (E-value threshold <0.05). Then we perform a BLASTn search using novel *wtf* genes from *S. octosporus*, *S. osmophilus,* and *S. cryophilus* as queries to find additional *wtf* genes and pseudogenes within each genome (E-value threshold <0.05; *Altschul et al., 1990*).

To search for *S. japonicus wtf* genes, we used sequences of *S. octosporus*, *S cryophilus,* and *S. pombe wtf* genes as query for BLAST with *S. japonicus* (*Altschul et al., 1990*; *Rhind et al., 2011*). This yielded no hits. We also carried out a MEME motif search of all the available *wtf* genes sequences and then perform PSI-BLAST to find genes with *wtf* genes motifs in *S. japonicus* (parameters: expect threshold: 10, word size: 3, matrix: BLOSUM62, gap costs = existence: 11, extension: 1, PSI-BLAST threshold: 0.005; *Altschul et al., 1997*). This also yielded no conclusive hits. Finally, we manually inspected *S. japonicus* genes defined as lineage-specific by OrthoVenn2 to search for multi-exon (5-6) candidate genes with a potential alternate translational start site in intron 1 or exon 2, similar to the *wtf* drivers (*Xu et al., 2019*). This search also yielded no promising hits.

## *S. octosporus wtf* and *wag* gene annotations

To annotate *wtf* genes in *S. octosporus* we used two different approaches listed below.

First, we aligned the short-read RNA-sequencing data described above to the *S. octosporus* genome using Geneious Prime 2021.1.1 (https://www.geneious.com). For each *wtf* and *wag* gene identified, we manually viewed RNA-sequencing data and used it to annotate the exons and introns. For genes and pseudogenes with insufficient sequence coverage, we determined coding sequences using homology to other *wtf* or *wag* genes that we were able to annotate with RNA-sequencing data. Specifically, we first aligned the unannotated genes with annotated genes using MAFFT with parameters L-INS-I (200PAM scoring matrix/k=2; Gap open penalty of 2; offset of 0.123; *Katoh et al., 2002*; *Katoh and Standley, 2013*). We then used the alignment to manually inspect genes to annotate splicing sites and predict coding sequences. Genes with incomplete coding sequences, including those determined to have lost splice sites, were considered pseudogenes.

Second, we further polished the annotation of *wtf* and *wag* genes according to the ONT cDNA sequencing data. We generated high-quality transcript information using the FLAIR pipeline as described above, and we predicted the longest open reading frame of these transcripts using Trans-Decoder (version: 5.5.0). We manually compared the coding frame of FLAIR transcripts mapped at the *wtf* or *wag* loci with the gene annotation obtained in the first approach and refined the gene structure annotation. Both pipeline of annotation gave similar results, we resolved manually the discrepancies between the different annotations and reported the annotation of *S. octosporus wtf* genes in *Supplementary file 1b*.

## *S. osmophilus wtf* and *wag* gene annotations

We first annotated *S. osmophilus wtf* and *wag* genes using Augustus prediction (trained with *S. octosporus* data; *Stanke et al., 2008*). We then manually inspected the annotations using alignments of all the *S. osmophilus wtf* or *wag* genes generated by MAFFT (L-INS-I; 200PAM scoring matrix/k=2; Gap open penalty of 2; offset of 0.123; *Katoh et al., 2002*; *Katoh and Standley, 2013*). Genes with four exons were annotated as pseudogenes when a fifth exon was not predicted by Augustus and was found to be absent after inspection of the alignment. In many of these pseudogenes, the fifth exons were degenerated with accumulated stop codons.

## 5S rDNA annotation in *S. octosporus,* *S. cryophilus,* and *S. osmophilus*

To annotate 5S rDNA in the genomes of *S. octosporus* and *S. cryophilus,*we used BLASTn using annotated 5S rDNA sequences from the corresponding genome as a query (*Altschul et al., 1990*). For *S. osmophilus*, we used an *S. octosporus* 5S rDNA gene as a query. In all genomes, hits with 70–100% DNA sequences identity were considered 5S rDNA genes. All 5S rDNA can be found in GFF files of annotated genomes (*Figure 1—source data 4–6*).

## LTR annotation in *S. osmophilus*

To annotate Tf transposon LTRs in *S. osmophilus*, we used BLASTn to search for sequences similar to the already annotated LTRs found in *S. cryophilus* and *S. pombe* (*Rhind et al., 2011*). We found many

hits in *S. osmophilus* (E-value less than 0.05). In addition, we also used the LTR_retriever program which identified additional LTRs in *S. osmophilus* (*Ou and Jiang, 2018*). All the LTRs identified are reported in *Supplementary file 1i*. *S. octosporus* does not have intact transposons or identified LTR sequences (*Rhind et al., 2011*).

## De novo discovery and genome-wide scanning of the FLEX motif (Mei4-binding motif)

To identify Mei4-binding motifs, we first compiled a list of 70 *S. pombe* Mei4 target genes. These genes were selected as Mei4 target genes based on the following criteria: (1) they were shown to contain Mei4 ChIP-seq peaks at two time points known to be the functionally critical period for Mei4 during meiosis (3 hr and 4 hr into meiosis; *Alves-Rodrigues et al., 2016*); and (2) they are among the genes expressed in the the middle of meiosis whose transcript levels were reduced in *mei4Δ* and increased when Mei4p was overexpressed (*Mata et al., 2007*). Among these 70 *S. pombe* Mei4 target genes, we further selected 49 genes that have single copy orthologs in other fission yeast species according to *Rhind et al., 2011* and the result of our orthovenn2 analysis (*Supplementary file 1f*). We extracted the 1000 bp sequences upstream of the start codon of these 49 genes in each species and performed de novo motif discovery using MEME (http://meme-suite.org/index.html; *Bailey and Elkan, 1994*). Manual inspection of all resulting motifs identified FLEX motifs in *S. pombe*, *S. octosporus*, *S. osmophilus,* and *S. cryophilus* (*Figure 2—figure supplement 2A*). We then combined all 196 genes in the four fission yeast species as input for de novo motif discovery, and the resulting 11 bp FLEX motif was submitted to the FIMO tool (*Grant et al., 2011*) for genome-wide motif scanning. A total of 33,089 FIMO hits were found in the 4 fission yeast species using the default p-value cutoff of 1E-4. By comparing the number of hits in Mei4 target genes and the number of hits in other genes, we chose p-value <3E-6 as the criterion for deeming a FIMO hit confident. This cutoff value maximized the number of hits in known *S. pombe* Mei4 target genes while minimizing the number of hits in genes not known to be Mei4 targets in *S. pombe*. In the reference *S. pombe* genome, there are a total of 476 FIMO hits meeting this criterion. Among the 49 *S. pombe* genes used for motif discovery, 59.2% (29 out of 49 genes) harbor at least one confident hit in the 1000 bp region upstream of the start codon, whereas for the other *S. pombe* genes, 6.5% (328 out of 5073 genes) harbor at least one confident hit in the 1000 bp region upstream of the start codon (p=7.47E-22, Fisher's exact test). The statistics of FIMO hits is shown in *Supplementary file 1g*, and all confident FIMO hits are listed in *Supplementary file 1h*.

## DNA sequence alignments and phylogenic tree construction

All DNA or amino acid sequence alignments were constructed using the MAFFT (*Katoh et al., 2002*; *Katoh and Standley, 2013*) plugin in Geneious Prime 2021.1.1 with parameters L-INS-I (200PAM scoring matrix/k=2; Gap open penalty of 2; offset of 0.123). We generated trees using the PhyML 3.0 (*Guindon et al., 2010*) in the webpage http://www.atgc-montpellier.fr/phyml/. The substitution model used was selected by smart model selection, which calculates an Akaike information criterion for each substitution model and then selects the best model for the dataset (*Akaike, 1998*; *Lefort et al., 2017*). The starting tree for each phylogeny was generated by BIONJ, an improved version of neighbor-joining (*Gascuel, 1997*). The trees were then improved with nearest neighbor interchange (*Felsenstein, 2004*). The branch support was calculated by Shimodaira-Hasegawa-like approximate likelihood ratio test (SH-like aLRT; *Anisimova and Gascuel, 2006*). Then the trees were rooted by midpoint using FigTree v1.4.4 (http://tree.bio.ed.ac.uk/software/figtree).

## Analyses of repetitive regions within *wtf* genes

We aligned the full length of all *S. octosporus*, *S. osmophilus wtf genes* within each species using MAFFT with parameters L-INS-I (200PAM scoring matrix/k=2; Gap open penalty of 2; offset of 0.123; *Katoh et al., 2002*; *Katoh and Standley, 2013*) using Geneious Prime 2021.1.1 (https://www.geneious.com). We then manually identified the repeat region within the alignments and manually quantified the number of bases within the repeat.

To obtain sequence logos of *S. octosporus* and *S. osmophilus* repeats in exon 4, we extracted the first complete repeat for all *wtf* genes containing a repeat. We then separately aligned all the *S. octosporus* and *S. osmophilus* repeats to produce FASTA files which we uploaded to the Weblogo3

interface (http://weblogo.threeplusone.com/; *Crooks et al., 2004*). The output generated the logos displayed in *Figure 5—figure supplement 3*.

## GARD analyses of recombination within *wtf* gene family

To study the recombination within *wtf* gene family within a species, we first produced an alignment of the coding sequence of *wtf* genes with translation align in Geneious Prime 2021.1.1 (https://www.geneious.com/) with MAFFT alignment L-INS-I (200PAM scoring matrix/k=2; Gap open penalty of 2; offset of 0.123; *Katoh et al., 2002*; *Katoh and Standley, 2013*). We then used our alignments to find recombination events within the *wtf* gene family by using GARD (*Kosakovsky Pond et al., 2006a*) with general discrete model of site-to-site variation with three class rates executed within the Datamonkey website (https://www.datamonkey.org/; *Weaver et al., 2018*).

## Syntenic analysis

To find *wtf* loci shared by *Schizosaccharomyces* species (*Figure 4* and *Figure 4—figure supplement 1*) and to assay the relationship between *wtf* loci and ancestral 5S rDNA sites (*Figure 6*), we manually inspected synteny of loci in *S. octosporus, S. osmophilus, S. cryophilus,* and *S. pombe*. In order to study the synteny between different *wtf* loci, we used OrthoVenn2 file generated previously (see *S. osmophilus* genome annotation section of Materials and methods) and the Ensembl fungi database to identify the orthologous genes (*Howe et al., 2019*; *Xu et al., 2019*). For each *wtf* locus we identified the immediately upstream and downstream gene and then the corresponding orthologs in each species. If the gene immediately upstream and/or downstream of the *wtf* loci did not correspond to any ortholog, we use the gene after and so on. An analogous approach was used with the analysis of 5S rDNA sites. All the data is reported in *Supplementary files 1l-m, p, and q*.

## *S. cerevisiae* LExA-ER-AD β-estradiol inducible system

The LExA-ER-AD system (*Ottoz et al., 2014*) uses a heterologous transcription factor containing LexA DNA-binding protein, the human estrogen receptor (ER), and an activation domain (AD). β-Estradiol binds the human ER and tightly regulates the activity of the LexA-ER-AD transcription factor. The LexA DNA-binding domain recognizes *lexA* boxes in the target promoter.

## Cloning *S. octosporus, S. osmophilus,* and *S. cryophilus wtf*^poison^ and *wtf*^antidote^ alleles for expression in *S. cerevisiae*

All plasmids used in this study are listed in *Supplementary file 3c*. All oligos used in this study are listed in *Supplementary file 3b*. All constructs were cloned into the KpnI or KpnI+BamHI sites of pRS314 or pRS316 (*Sikorski and Hieter, 1989*).

### Cloning *S. octosporus wtf61*^poison^ (SOCG_04114) under the control of *a β*-estradiol inducible promoter (pSZB1040)

We amplified the predicted coding sequence of the *S. octosporus wtf61*^poison^ from a gBlock synthetized by IDT (Coralville, IA) via PCR using oligos 1432 and 1442. The *CYC1* terminator was digested from pSZB395 (*Nuckolls et al., 2020*) using SfiI and XhoI. We then cloned *S. octosporus wtf61*^poison^ CDS and the CYC1 terminator into XhoI and BamHI site of pSZB385 to generate SZB985. We then digested pSZB985 with XhoI and BamHI to extract *wtf61*^poison^ CDS with the CYC1 terminator. We next PCR amplified the LexA promoter (LexApr) using oligos 1195 and 1240 from FRP1642 (Addgene #58442; *Ottoz et al., 2014*). We then cloned both the promoter and the *wtf61*^poison^ CDS fragment into pRS316 (*Sikorski and Hieter, 1989*).

### Cloning *S. octosporus wtf61*^antidote^ (SOCG_04114) under the control of *a β*-estradiol inducible promoter (pSZB1108)

We amplified the predicted *S. octosporus wtf61*^antidote^ from a gBlock synthetized by IDT (Coralville, IA) via PCR using oligos 2011 and 2170. We PCR amplified CYC1 terminator from pSZB1040 using oligos 2194 and 2195. We then used overlap PCR to stitch together *S. octosporus wtf61*^antidote^ and *CYC1* terminator via PCR using oligos 2011 and 2195. We digested LexApr (described above for pSZB1140) with KpnI and XhoI. We digested the fragment *S. octosporus wtf61*^antidote^-*CYC1* with XhoI and BamHI.

Finally, we cloned LexApr and *S. octosporus wtf61*<sup>antidote</sup>-*CYC1* fragments into pRS314 (*Sikorski and Hieter, 1989*).

### Cloning *S. cryophilus wtf1*<sup>poison</sup> (SPOG_03611) under the control of *a β-estra-diol* inducible promoter (pSZB1122)

We amplified the predicted coding sequence of the *S. cryophilus wtf1*<sup>poison</sup> (SPOG_03611) from a gBlock synthetized by IDT (Coralville, IA) via PCR using oligos 2277 and 2278. We amplified CYC1 terminator from pSZB1040 using oligos 2279 and 2170. We used overlap PCR to stitch together *S. cryophilus wtf1*<sup>poison</sup> with *CYC1* terminator using oligos 2277 and 2170. We digested that PCR product with XhoI and BamHI. We then cloned the *S. cryophilus wtf1*<sup>poison</sup>-*CYC1* and LexApr (described above for pSZB1140) cassettes into of pRS316 (*Sikorski and Hieter, 1989*).

### Cloning *S. cryophilus wtf1*<sup>antidote</sup> (SPOG_03611) under the control of *a* β-estra-diol inducible promoter (pSZB1192)

We amplified the predicted coding sequence of *S. cryophilus wtf1*<sup>antidote</sup> from a gBlock synthetized by IDT (Coralville, IA) via PCR using oligos 2276 and 2278. We used overlap PCR to stitch together *S. cryophilus wtf1*<sup>antidote</sup> and *CYC1* terminator (described above for pSZB1122) using oligos 2276 and 2170. We then digested the resulting PCR product with XhoI and BamHI. We then cloned both the LexApr (described above for pSZB1140) and *S. cryophilus wtf1*<sup>antidote</sup>-*CYC1* fragments into pRS314 (*Sikorski and Hieter, 1989*).

### Cloning *S. osmophilus wtf41*<sup>poison</sup> under the control of *a* β-estradiol inducible promoter (pSZB1327)

We amplified the predicted coding sequence of *S. osmophilus wtf41*<sup>poison</sup> from a gBlock synthetized by IDT (Coralville, IA) via PCR using oligos 2783 and 2780. We amplified the *CYC1* terminator from pSZB1040 via PCR using oligos 2781 and 2771. We amplified the LexApr from pSZB1040 via PCR using oligos 1195 and 2778. We used overlap PCR to stitch together LexApr, *S. osmophilus wtf41*<sup>poison</sup>, and the *CYC1* terminator using oligos 1195 and 2771. We then cloned the resulting product into pRS316 (*Sikorski and Hieter, 1989*).

### Cloning *S. osmophilus wtf41*<sup>antidote</sup> under the control of *a β-estradiol* induc-ible promoter (pSZB1325)

We amplified the predicted coding sequence of *S. osmophilus wtf41*<sup>antidote</sup> from a gBlock synthetized by IDT (Coralville, IA) via PCR using oligos 2779 and 2780. We amplified LexApr from pSZB1040 via PCR using oligos 1195 and 2782. We use overlap PCR to stitch together the LexApr, *wtf41*<sup>antidote</sup>, and the *CYC1* terminator (described above for pSZB1327) using oligos 1195 and 2771. We then cloned the product into pRS314 (*Sikorski and Hieter, 1989*).

### Cloning *S. octosporus wtf25*<sup>poison</sup> (SOCG_04480)-GFP under the control of *a* β-estradiol inducible promoter (pSZB1353)

We amplified the predicted coding sequence of *S. octosporus wtf25*<sup>poison</sup> from a gBlock synthetized by IDT (Coralville, IA) via PCR using oligos 2669 and 2830. We amplified LexApr from SZB1040 via PCR using oligos 1195 and 2668. We amplified GFP from pKT0127 (*Sheff and Thorn, 2004*) via PCR using oligos 2831 and 2832. We amplified the *CYC1* terminator from SZB1040 using oligos 2833 and 2771. We used overlap PCR to stitch together LexApr-*S. octosporus wtf25*<sup>poison</sup>-GFP-*CYC1* terminator using oligos 1195 and 2771. We then cloned the digested product into pRS316 (*Sikorski and Hieter, 1989*).

### Cloning *S. octosporus wtf25*<sup>antidote</sup> (SOCG_04480) mCherry under the control of *a* β-estradiol inducible promoter (pSZB1347)

We amplified the predicted coding sequence of *S. octosporus wtf25*<sup>antidote</sup> from a gBlock synthetized by IDT (Coralville, IA) via PCR using oligos 2662 and 2663. We amplified LexApr from pSZB1040 via PCR using oligos 1195 and 2661. We amplified mCherry from pSZB457 via PCR using oligos 2664 and 2665. We amplified *CYC1* terminator from pSZB1040 via PCR using oligos 2666 and 2771. We used

overlap PCR to stitch together the three products. We then cloned the resulting KpnI-digested PCR product into pRS314 (*Sikorski and Hieter, 1989*).

### Cloning *S. osmophilus wtf19*poison under the control of *a* β-estradiol inducible promoter (pSZB1324)

We amplified the predicted coding sequence of *S. osmophilus wtf19*poison from a gBlock synthetized by IDT (Coralville, IA) via PCR using oligos 2777 and 2774. We amplified LexApr from pSZB1040 via PCR using oligos 1195 and 2776. We amplified *CYC1* terminator from pSZB1040 via PCR using oligos 2775 and 2771. We use overlap PCR to stitch together LexApr-*S. osmophilus wtf19*poison-mCherry-*CYC1* terminator using oligos 1195 and 2771. We cloned LexApr-*S. osmophilus wtf19*poison -*CYC1* terminator into pRS316 (*Sikorski and Hieter, 1989*).

### Cloning *S. osmophilus wtf19*poison under the control of *a* β-estradiol inducible promoter (pSZB1322)

We amplified the predicted coding sequence of *S. osmophilus wtf19*antidote from a gBlock synthetized by IDT (Coralville, IA) via PCR using oligos 2773 and 2774. We amplified LexApr from pSZB1040 via PCR using oligos 1195 and 2772. We used overlap PCR to stitch together LexApr-*S. osmophilus wtf19*antidote-*CYC1* terminator (described above for pSZB1324) using oligos 1195 and 2771. We cloned LexApr-*S. osmophilus wtf19*poison- *CYC1* terminator into pRS314 (*Sikorski and Hieter, 1989*).

## Plasmid transformation in *S. cerevisiae*

All yeast strains used in this study are listed in *Supplementary file 3a* with detailed genotype and citation information. Plasmids used in this study are listed in *Supplementary file 3c*. We transformed plasmids into *S. cerevisiae* SZY1637 (*Nuckolls et al., 2020*) using a protocol modified from *Elble, 1992*. Specifically, we incubated a yeast colony in a mix of 240 µL 50% PEG3500, 36 µL 1 M lithium acetate, 50 µL boiled salmon sperm DNA (10 mg/ml), and 10 µL plasmid for 4–6 hr at 30°C before selecting transformants. We selected transformants on synthetic complete (SC) media (6.7 g/L yeast nitrogen base without amino acids and with ammonium sulfate, 2% agar, 1 X amino acid mix, 2% glucose) lacking histidine, uracil, and tryptophan (SC -His -Ura -Trp).

## Spot assays in *S. cerevisiae*

We grew 5 mL overnight cultures in SC -His -Ura -Trp of each strain. We then diluted each culture to an $OD_{600}$ of 1 and performed a serial dilution. We then plated 10 µL of each dilution on a solid SC -His -Ura -Trp petri plate with or without 500 nM β-estradiol.

## Imaging Wtf proteins expressed in *S. cerevisiae*

For imaging of Wtf proteins expressed in *S. cerevisiae* (*Figure 7D–F*), we first grew 5 mL saturated overnight cultures in SC -His -Ura -Trp media. The next day, we diluted 1 mL of each saturated culture into 4 mL of fresh SC -His -Ura -Trp media. We then added β-estradiol to a final concentration of 500 nM to induce *wtf* expression and shook the cultures at 30°C for 4 hr prior to imaging.

Cells (5 µL concentrated culture) were then imaged on an LSM-780 (Zeiss) with a ×40 LD C-Apochromat (NA = 1.1) objective. A physical zoom of 8 was used which yielded an XY pixel size of 0.052 µm. The fluorescence of GFP was excited with the 488 nm laser and filtered through a 491–553 nm bandpass filter before being collected onto a GaAsP detector running in photon counting mode. The fluorescence of mCherry was excited with the 561 nm laser and filtered through a 562–624 nm bandpass filter before being collected onto the same detector.

## *S. octosporus* strains

The two wild-type heterothallic *S. octosporus* strains (DY44286=NIG10005 and DY44284=NIG10006) were a kind gift from Dr. Hironori Niki, and all other *S. octosporus* strains were constructed based on these two heterothallic strains. *S. octosporus*-related genetic methods are performed according to or adapted from genetic methods for *S. pombe* (*Forsburg and Rhind, 2006*; *Seike and Niki, 2017*). The construction of *wtf* gene deletion strains was carried out by PCR-based gene targeting using an SV40-EM7 (SVEM) promoter-containing G418-resistance marker referred to here as *kanSVEM* (*Erler et al.,*

*2006*). As sequences between the *wtf*-flanking 5S rDNA genes share high similarity among different *wtf* gene loci, to ensure the specificity of gene deletion, we used homologous arm sequences outside of 5S rDNA genes, and the length of at least one homologous arm was above 1 kb. All *wtf* gene deletion strains were verified using PCR. PCR primer sequences are listed in *Supplementary file 3b*.

To analyze the spore-killing activity of *wtf25* at an ectopic genomic locus, we constructed integrating plasmids based on the pDB4978 vector described below. A pDB4978-based plasmid was linearized with NotI digestion and integrated at the *leu1* (*SOCG_02003*) locus. Transformants were selected by the resistance to clonNAT conferred by the natMX marker on pDB4978. Successful integration resulted in the deletion of the ORF sequence of the *leu1* (*SOCG_02003*) gene and leucine auxotrophic phenotype (*Figure 10*).

## Integration plasmids for *S. octosporus*

All *S. octosporus* plasmids were generated by recombination cloning using the ClonExpressII One Step Cloning Kit (Vazyme, Nanjing, China). For the construction of the pDB4978 vector, the plasmid pAV0584 (*Vještica et al., 2019*) was first digested using NotI and HindIII, and the largest resulting fragment (about 4.5 kb) was purified and then digested using SpeI to obtain an approximately 3.7 kb fragment containing AmpR, ori, and the natMX marker. A sequence containing the f1ori and multiple cloning sites was PCR amplified from pAV0584 using primers oGS-177 and oGS-178 (oligo sequences are listed in *Supplementary file 3b*). The sequences upstream and downstream of the *leu1(SOCG_02003)* ORF were amplified from *S. octosporus* genomic DNA using primers oGS-192 and oGS-193, and primers oGS-195 and oGS-197, respectively. Finally, all four fragments were combined by recombination cloning to generate the pDB4978 vector.

## Spore viability analysis

Spore viability was assessed by octad dissection using a TDM50 tetrad dissection microscope (Micro Video Instruments, Avon, USA). The method of octad dissection was adapted from *Seike and Niki, 2017*, and a detailed description of the experiment procedure follows. First, to maximize mating efficiency, before mating, all parental strains were streaked on YES plates for overnight growth. Then, parental strains were mixed at a one-to-one ratio and dropped on PMG plate (or PMG plates with the leucine supplement for leucine auxotrophic strains) and incubated at 30°C. After 2 days, about 1 $OD_{600}$ unit of cells were resuspended in 200 µl of 1 mg/ml solution of snailase (Beijing Solarbio Science & Technology Co.). The mixture was incubated without agitation at 25°C for 1 day and then the supernatant was aspirated. Snailase-treated cells were diluted in sterile water and then dropped on a YES plate for octad dissection. After dissection, plates were incubated at 30°C for about 5 days, and then plates were scanned, and the genotypes of colonies were determined by replica plating.

For data analysis, we excluded spores dissected from asci with fewer than eight spores (asci with fewer than eight spores are rare when sporulation was conducted on PMG plates) and octads containing greater than four spores harboring one allele of a heterozygous locus (excluded octads represent <2% of the octads analyzed). Numeric data of octad dissection analysis are in *Figure 8—source data 1*; *Figure 9—figure supplements 1–6*; *Figure 9—source data 1* and the scanned plate photos are in the; *Figure 9—source data 2*; *Figure 9—figure supplements 1–6*; *Figure 9—source data 2 and 3*. For statistical analysis of the spore viability data, Fisher's exact test was performed using the web page https://www.langsrud.com/fisher.htm, and exact binomial test was performed using an Excel spreadsheet downloaded from http://www.biostathandbook.com/exactgof.html (*McDonald, 2009*).

## Acknowledgements

We thank members of the Zanders lab for their helpful comments on the paper. We are grateful to Dr. Hironori Niki for sharing strains and Dr. Taisuke Seike for helpful advice. The original data underlying this manuscript can be accessed from the Stowers Original Data Repository at ftp://odr.stowers.org/LIBPB-1744. This work was supported by grants from the Chinese Ministry of Science and Technology and the Beijing municipal government to (L-LD), the Stowers Institute for Medical Research (SEZ), the Searle Scholars Award (SEZ), and the National Institutes of Health (NIH) DP2GM132936 (SEZ). The funders had no role in study design, data collection and analysis, or manuscript preparation. The

content is solely the responsibility of the authors and does not necessarily represent the official views of the funders.

## Additional information

### Funding

| Funder | Grant reference number | Author |
|---|---|---|
| NIH Office of the Director | DP2GM132936 | Sarah E Zanders |
| Stowers Institute for Medical Research | | Sarah E Zanders |
| Kinship Foundation | Searle Scholars Award | Sarah E Zanders |
| Chinese Ministry of Science and Technology | | Li-Lin Du |
| Beijing municipal government | | Li-Lin Du |

The funders had no role in study design, data collection and interpretation, or the decision to submit the work for publication.

### Author contributions

Mickaël De Carvalho, Guo-Song Jia, Conceptualization, Data curation, Formal analysis, Validation, Investigation, Visualization, Methodology, Writing – original draft, Writing – review and editing; Ananya Nidamangala Srinivasa, Data curation, Formal analysis, Validation, Investigation, Visualization, Writing – review and editing; R Blake Billmyre, Formal analysis, Validation, Investigation, Visualization, Methodology, Writing – review and editing; Yan-Hui Xu, Data curation, Formal analysis, Validation, Investigation, Visualization; Jeffrey J Lange, Data curation, Formal analysis, Validation, Investigation, Visualization, Methodology, Writing – review and editing; Ibrahim M Sabbarini, Formal analysis, Validation, Investigation, Methodology, Writing – review and editing; Li-Lin Du, Conceptualization, Data curation, Formal analysis, Supervision, Funding acquisition, Validation, Investigation, Visualization, Methodology, Writing – original draft, Project administration, Writing – review and editing; Sarah E Zanders, Conceptualization, Data curation, Supervision, Funding acquisition, Investigation, Visualization, Methodology, Writing – original draft, Project administration, Writing – review and editing

### Author ORCIDs

Guo-Song Jia http://orcid.org/0000-0002-8731-8606
Ananya Nidamangala Srinivasa http://orcid.org/0000-0003-1487-2793
R Blake Billmyre http://orcid.org/0000-0003-4866-3711
Jeffrey J Lange http://orcid.org/0000-0003-4970-6269
Li-Lin Du http://orcid.org/0000-0002-1028-7397
Sarah E Zanders http://orcid.org/0000-0003-1867-986X

### Decision letter and Author response

Decision letter https://doi.org/10.7554/eLife.81149.sa1
Author response https://doi.org/10.7554/eLife.81149.sa2

## Additional files

### Supplementary files

• Supplementary file 1. Annotations, regulatory motifs, conservation, and genomic context of *wtf* genes. (a) Percent amino acid identity of all 1:1 orthologs in *Schizosaccharomyces*. Orthologous gene sets between pairs of *Schizosaccharomyces* species were identified using a combination of Orthovenn2 and BLASTp (*Xu et al., 2019*). All proteins from a given species were aligned the proteins of the other species, and the best hit for each was used to determine the amino acid identity. All the percent identity values between a pair of species were then used to calculate the average amino acid identity between the two species. The genome used for finding proteins

sequences was generated by *Rhind et al., 2011* for *Schizosaccharomyces octosporus*, *S. cryophilus*, *S. pombe*, and *S. japonicus*. The *S. osmophilus* genome was sequenced and annotated in this study (see Materials and methods). The orthologs list can be found in *Figure 1—source data 3*. (b) Location and features of *S. octosporus wtf* genes. *S. octosporus wtf* genes names are found in column A. The gene locations are described from columns B to F. If the gene is associated with a *wag* gene, the *wag* gene name and orientation are indicated in columns G and H. Column K indicates whether the *wtf* gene is associated with a 5S rDNA gene (immediately adjacent to the *wtf* or outside a flanking *wag* gene). The strand location of 5S rDNA genes that may be found upstream of the *wtf* gene is described in column I, while the strand location for 5S rDNA genes that may be downstream of the *wtf* gene is described in column J. *wtf* genes and the associated 5S rDNA are considered to be in tandem when they are encoded in the same strand and in the same direction. The *wtf* and *wag* genes are all in a divergent orientation in that they are on opposite strands and transcribed in opposite directions. Column L details if there is a 5S rDNA upstream, downstream or if there is a 5S rDNA gene both upstream and downstream the *wtf* gene. Column M describes our prediction if the *wtf* gene encodes a driver (intact poison start codon), an antidote-only gene (no start codon for poison), or is a pseudogene (premature stop codon). Columns N and O show the read counts of the two isoforms detected with long read RNA-seq, respectively, with the long isoform predicted to encode an antidote protein and the short isoform predicted to encode a poison protein. Column P indicates if a FIMO motif scanning hit of the FLEX motif was present in intron 1 of the *wtf* gene. Column Q provides the location of the FIMO hit in intron 1 (only the best scoring FIMO hit is shown if more than one hit was found). Column R shows the strand the FIMO hit is on. Columns S and T show the p-value of the FIMO hit and the sequence of the FIMO hit, respectively. (c) Location and features of *S. osmophilus wtf* genes. *S. osmophilus wtf* genes names are found in column A. The gene locations are described from columns B to F. If the gene is associated with a *wag* gene, the *wag* gene name and orientation are indicated in columns G and H. Column K indicates whether the *wtf* gene is associated with a 5S rDNA gene (immediately adjacent to the *wtf* or outside a flanking *wag* gene). The strand location of 5S rDNA genes that may be found upstream of the *wtf* gene is described in column I, while the strand location for 5S rDNA genes that may be downstream of the *wtf* gene is described in column J. *wtf* genes and the associated 5S rDNA are in tandem when they are encoded in the same strand and in the same direction. Column L details if there is a 5S rDNA upstream, downstream or if there is a 5S rDNA gene both upstream and downstream the *wtf* gene. Column M describes our prediction if the *wtf* gene encodes a driver (intact poison start codon), an antidote-only gene (no start codon for poison), or is a pseudogene (premature stop codon). Columns N and O indicated the strand of the LTR and orientation relative to the *wtf* gene. As above, tandem orientation means same orientation and same strand, convergent means the elements are on opposite strands but are transcribed toward each other. Divergent means that the elements are in different strands and are transcribed in opposite directions. Column P indicates if a FIMO motif scanning hit of the FLEX motif was present in intron 1 of the *wtf* gene. Column Q provides the location of the FIMO hit in intron 1 (only the best scoring FIMO hit is shown if more than one hit was found). Column R shows the strand the FIMO hit is on. Columns S and T show the p-value of the FIMO hit and the sequence of the FIMO hit, respectively. (d) Location and features of *S. cryophilus wtf* genes. *S. cryophilus wtf* genes names are found in column A. The gene locations are described from columns B to F. If the gene is associated with a *wag* gene, the *wag* gene name and orientation are indicated in columns G and H. Column K indicates whether the *wtf* gene is associated with a 5S rDNA gene (immediately adjacent to the *wtf* or outside a flanking *wag* gene). The strand location of 5S rDNA genes that may be found upstream of the *wtf* gene is described in column I, while the strand location for 5S rDNA genes that may be downstream of the *wtf* gene is described in column J. *wtf* genes and the associated 5S rDNA are in tandem (column L) when they are encoded in the same strand and in the same direction. Column L details if there is a 5S rDNA upstream, downstream, or if there is a 5S rDNA gene both upstream and downstream the *wtf* gene. Column M describes our prediction if the *wtf* gene encodes a driver (intact poison start codon), an antidote-only gene (no start codon for poison), or is a pseudogene (premature stop codon). Column N indicates if a FIMO motif scanning hit of the FLEX motif was present in intron 1 of the *wtf* gene. Column O provides the location of the FIMO hit in intron 1 (only the best scoring FIMO hit is shown if more than one hit was found). Column P shows the strand the FIMO hit is on. Columns Q and R show the p-value of the FIMO hit and the sequence of the FIMO hit, respectively. (e) Pairwise amino acid identity of intact *wtf* genes. Using MAFFT with parameters L-INS-I (200PAM scoring matrix/k=2; Gap open penalty of 2; offset of 0.123), we aligned all the predicted coding sequences of the intact *wtf* genes from *S. octosporus, S. osmophilus, S. cryophilus,* and *S. pombe*. The longest isoform (i.e. antidote) of each protein, when two isoforms are predicted, was used. The

table shows the percent amino acid identity shared between all pairs of genes. The cells are color-coded such that pairs with higher similarity are shaded a darker red. (f) Genes used for FLEX motif discovery. This table lists the 49 *S. pombe* Mei4 target genes and their orthologs in three other fission yeast species used for FLEX motif discovery. (g) Summary statistics of genome-wide FLEX motif scanning. FIMO hits were classified into unreliable hits and confident hits using the p-value cutoff of 3E-6. This table lists the numbers of total FIMO hits, unreliable hits, and confident hits in each species. (h) Confident hits of FLEX motif scanning. This table lists the confident FIMO hits in the four fission yeast species. (i) Locations of LTR sequences in *S. osmophilus*. We used BLASTn with *S. cryophilus* LTR sequences as queries to identify *S. osmophilus* LTRs. In addition, we also used as LTR_retriever (see Materials and methods). The table reports the location, length, and orientation of each LTR identified. (j) Summary of association between 5S rDNA and *wtf* genes within *Schizosaccharomyces* genomes. The table lists the number of 5S rDNA genes in each species and details how many of those 5S rDNA genes are associated with a locus that contains one or more *wtf* genes. Additional unannotated 5S rDNA genes were identified within the *S. octosporus* and *S. cryophilus* genomes using BLASTn. In *S. osmophilus*, all 5S rDNA genes were identified by BLASTn. A gene was considered a *bona fide* 5S rDNA gene if it shared more than 70% sequence identity with another 5S rDNA gene in that genome. A 5S rDNA was considered associated with a *wtf* locus if it was immediately adjacent to a *wtf* gene, or if it was adjacent to a *wag* gene flanking a *wtf* gene. (k) *wag* gene transcripts in *S. octosporus*. Annotation of *wag* genes of *S. octosporus* with the corresponding SOCG names, where applicable, in column B. Genes with early stop codons relative to consensus sequences are considered pseudogenes (column H). (l) Synteny analysis of the regions containing *wtf* genes in *S. pombe* (i.e. *Figure 4* and *Figure 4—figure supplement 1*). For each *S. pombe wtf* locus (from the *S. kambucha* isolate; column A), we noted the genes directly upstream and downstream excluding *wag* genes (columns H and I). We next found the orthologs of those *wtf*-flanking genes in *S. osmophilus* (columns J and K), *S. octosporus* (columns L and M), and *S. cryophilus* (columns N and O). If the orthologs of the genes that flank a *wtf* in *S. pombe* also flank a single *wtf* locus in another species, the *wtf* genes were considered to share 'complete' synteny. If the orthologs both flank *wtf* genes, but not the same *wtf* gene in a different species, we dubbed this scenario 'double partial synteny'. If only one of the two orthologs flanks a *wtf* gene in another species, we considered that 'partial synteny'. The synteny analyses results for *S. cryophilus*, *S. octosporus,* and *S. osmophilus* are reported in columns B-C, D-E, and F-G, respectively. (m) *S. cryophilus wtf* genes in synteny with *S. octosporus*, *S. osmophilus*, and *S. pombe wtf* genes (*Figure 4—figure supplement 1*). For each *S. cryophilus wtf* gene (column A), we noted the genes directly upstream and downstream, excluding *wag* genes (columns H and I). We next found the orthologs of those *wtf*-flanking genes in *S. octosporus* (columns J and K), *S. pombe* (columns L and M), and *S. osmophilus* (columns N and O). If the orthologs both flank *wtf* genes, but not the same *wtf* gene in a different species, we dubbed this scenario 'double partial synteny'. If only one of the two orthologs flanks a *wtf* gene in another species, we considered that 'partial synteny'. The synteny analyses results for *S. octosporus*, *S. pombe*, and *S. osmophilus* are reported in columns B-C, D-E, and F-G, respectively. (n) Percent amino acid identity of genes flanking *wtf* genes at syntenic loci (i.e. *Figure 4* and *Figure 4—figure supplement 1*). The amino acid sequences of genes flanking the *S. pombe wtf* loci shown in *Figure 4* (*wtf34*) and *Figure 4—figure supplement 1* (*wtf6*) were aligned with their orthologs from all other *Schizosaccharomyces* species using MAFFT L-INS-I (200PAM scoring matrix/k=2; Gap open penalty of 2; offset of 0.123). The tables depict the pairwise percent amino acid identity between all ortholog pairs. Comparisons between the genes flanking *S. pombe wtf34* (*clr4* and *met17*) are shown at the top, while the comparisons between the genes flanking *S. pombe wtf6* (*ago1* and *cyp9*) are shown below. (o) Species-specific *wtf* genes. Summary of the species-specific *wtf* loci and genes found in each species. The *S. kambucha* isolate of *S. pombe* was used for this table, and the reference genomes were used for the other species. The gene names of the species-specific *wtf* genes are shown in the final column. Genes found at separate loci are separated by commas and genes found at a centromere are shown in bold. (p) Analyzing if 5S rDNA genes are found at loci syntenic to 5S rDNA-adjacent *S. osmophilus wtf* genes in other species (i.e. *Figure 6*). For each *S. osmophilus wtf* locus (column A), we noted the genes directly upstream and downstream (columns D and E) excluding any *wag* genes. We next found the orthologs of those *wtf*-flanking genes in *S. octosporus* (columns F and G), and *S. cryophilus* (columns H and I). The synteny analyses results comparing *S. osmophilus wtf* loci to *S. octosporus* are shown in columns B and C. If the orthologs of the genes that flank a *wtf* in *S. osmophilus* also flank a single *wtf* locus in the queried species, the *wtf* genes were considered to share 'complete' synteny. If the orthologs both flank a *wtf* locus but not the same *wtf* locus in the queried species, we dubbed this scenario 'double partial synteny'. If only one of the two orthologs flanks a *wtf* gene in the queried species, we

considered that 'partial synteny'. For the analysis, we considered loci in complete synteny where there was a *wtf* gene flanked by a 5S rDNA gene in *S. osmophilus* (column J), but no *wtf* gene at the syntenic locus in the queried species (columns K and M, respectively). We evaluated if the *wtf*-lacking syntenic locus in *S. octosporus* or *S. cryophilus* contained a 5S rDNA gene (columns L and N, respectively). The loci that met our criteria and were considered in the analysis are listed in columns O and P for *S. octosporus* and *S. cryophilus*, respectively. In column Q, we considered each locus to be a lineage-specific locus meaning no synteny found in other species. (q) Analyzing if 5S rDNA genes are found at loci syntenic to 5S rDNA-adjacent *S. octosporus wtf* genes in other species (i.e. *Figure 6*). For each *S. octosporus wtf* locus (column A), we noted the genes directly upstream and downstream (columns D and E) excluding any *wag* genes. We next found the orthologs of those *wtf*-flanking genes in *S. osmophilus* (columns F and G) and *S. cryophilus* (columns H and I). The synteny analyses results comparing *S. octosporus wtf* loci to *S. osmophilus* are shown in columns B and C. If the orthologs of the genes that flank a *wtf* in *S. octosporus* also flank a single *wtf* locus in the queried species, the *wtf* genes were considered to share 'complete' synteny. If the orthologs both flank a *wtf* locus but not the same *wtf* locus in the queried species, we dubbed this scenario 'double partial synteny'. If only one of the two orthologs flanks a *wtf* gene in the queried species, we considered that 'partial synteny'. For the analysis, we considered loci in complete synteny where there was a *wtf* gene flanked by a 5S rDNA gene in *S. octosporus* (column J), but no *wtf* gene at the syntenic locus in the queried species (columns K and M, respectively). We evaluated if the *wtf*-lacking syntenic locus in *S. osmophilus* or *S. cryophilus* contained a 5S rDNA gene (columns L and N, respectively). The loci that met our criteria and were considered in the analysis are listed in columns O and P for *S. osmophilus* and *S. cryophilus*, respectively. In column Q, we considered each locus to be a lineage-specific locus meaning no synteny found in other species. (r) Repeat count within exon 4 in *S. octosporus* and *S. osmophilus wtf* genes (i.e. *Figure 5—figure supplement 2*). This tab contains 4 tables. From left to right, the first table displays the size, in base pairs of the repeat region found in each intact *S. octosporus wtf* genes. These sizes were determined manually in each gene. The next table summarizes how many *S. octosporus wtf* genes were found with repeat regions of the indicated ranges. The following two tables repeat the analyses with the *S. osmophilus wtf* genes. (s) Expanded analysis of *wtf*+5S rDNA loci in species A with 5S rDNA at the locus in species B (i.e. *Figure 6*). Expanded table of data presented in *Figure 6*. The analysis considers *wtf*+5S rDNA loci that are present in species A that are not found in species B. The total number of such sites, in addition to how many of the sites have a 5S rDNA gene at the syntenic site in species B is reported. The *wtf* genes considered are shown in the last column. Those with a 5S rDNA gene at the syntenic site in species B are shown in bold. Genes found at separate loci are separated by commas.

• Supplementary file 2. *S. octosporus* spore dissection analyses. (a) Total viability numerical data summary. (b) *wtf25*(SOCG_04480) deletion related numerical data of the octad dissection analysis. (c) *wtf68*(SOCG_01240) deletion related numerical data of the octad dissection analysis. (d) *wtf33* deletion related numerical data of the octad dissection analysis. (e) *wtf46*(SOCG_00084) deletion related numerical data of the octad dissection analysis. (f) *wtf60*(SOCG_04742) deletion related numerical data of the octad dissection analysis. (g) *wtf62*(SOCG_04077) deletion related numerical data of the octad dissection analysis. (h) *wtf21*(SOCG_02322) deletion related numerical data of the octad dissection analysis. (i) octo-pSIV-leu1-1D plasmid related numerical data of the octad spore dissection analysis.

• Supplementary file 3. Summary of yeast strains, plasmids and oligos. (a) Yeast strain summary. (b) Oligos summary. (c) Plasmids summary.

• MDAR checklist

### Data availability

*S. osmophilus* genomic sequencing data were deposited on the SRA under project accession code PRJNA839783. *S. octosporus* RNA sequencing data were deposited at NCBI SRA under the accession numbers SRR17543072 and SRR17543073.

The following datasets were generated:

| Author(s) | Year | Dataset title | Dataset URL | Database and Identifier |
|---|---|---|---|---|
| Billmyre RB, DeCarvalho M, Zanders S | 2022 | Schizosaccharomyces osmophilus long read sequencing | https://www.ncbi.nlm.nih.gov/bioproject/PRJNA839783 | NCBI BioProject, PRJNA839783 |
| Du L-L | 2022 | RNA sequencing and Nanopore cDNA sequencing of Schizosaccharomyces octosporus diploid cells undergoing azygotic meiosis | https://www.ncbi.nlm.nih.gov/bioproject/PRJNA795833 | NCBI BioProject, PRJNA795833 |
| Jia G-S, Du L-L | 2022 | S. octosporus RNA sequencing data were deposited at NCBI SRA under the accession numbers SRR17543072 | https://www.ncbi.nlm.nih.gov/geo/query/acc.cgi?acc=SRR17543072 | NCBI Gene Expression Omnibus, SRR17543072 |
| Jia G-S, Du L-L | 2022 | S. octosporus RNA sequencing data were deposited at NCBI SRA under the accession numbers SRR17543073 | https://www.ncbi.nlm.nih.gov/geo/query/acc.cgi?acc=SRR17543073 | NCBI Gene Expression Omnibus, SRR17543073 |

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
