## [Editor Report]

This paper presents important findings on the long-term evolutionary persistence of a meiotic driver gene family across several species of fission yeasts. The authors provide compelling evidence from phylogenetic analyses, comparative genomics, and functional experiments that *wtf* genes have an ancient origin in *Schizosaccharomyces* and retain the ability to drive. Based on their finding of extensive gene duplication and gene conversion throughout the evolutionary history of *wtf* genes, the authors also present an interesting hypothesis to explain how the ability to drive might be maintained over long evolutionary timescales – namely, that it is a property of the gene family as a whole, rather than of a single locus. Whether such a scenario is unique to fission yeasts or applies more broadly to other taxa is currently unknown, but this work certainly represents one of the most detailed mechanistic studies of selfish genes in any wild species to date, and thus provides a valuable example for future studies of how such systems might evolve.

---

## [Decision Letter]

**Decision letter after peer review:**

Thank you for submitting your article "The *wtf* meiotic driver gene family has unexpectedly persisted for over 100 million years" for consideration by *eLife*. Your article has been reviewed by 3 peer reviewers, including Andrea Sweigart as Reviewing Editor and Reviewer #1, and the evaluation has been overseen by Molly Przeworski as the Senior Editor.

Essential revisions:

1) To ensure your conclusions are bounded by the results, replace your verbal model with a discussion of newly discovered mechanisms of wtf evolution (e.g., expansion, gene conversion). In this discussion, take care to acknowledge that questions remain about the timescale of wtf fixation relative to the emergence of new drivers.

2) Please temper your claims in the Introduction and Discussion sections related to finding evidence of ACTIVE drive across the evolutionary history of this group.

3) Temper your claims about protein localization patterns and functional conservation.

4) Adjust your wording to avoid seeming dismissive of additional (non-drive) functions for wtf genes. Also, rewrite to clarify their role as suppressors.

*Reviewer #2 (Recommendations for the authors):*

The "model".

My sense is the paper would be improved by replacing the "model" with a brief sentence about, the diverse evolutionary mechanisms discovered here, which likely play a key role in the existence of wtf-based drive in the long-diverged taxa found here.

Does wtf somehow otherwise impact fitness.

The authors present and disregard an additional potential part of the evolution of wtf drivers… "… [wtf-driver] genes are not merely selfish parasites… [but may] promote fitness in some way that [they] have yet to discover. This theoretical additional function of the wtf genes could have promoted their long-term maintenance in fission yeast genomes. It is important to note, however, that genes do not need to promote fitness to be maintained in genomes and there is currently no evidence supporting a role of wtf drivers in promoting fitness, except in cases where they suppress other wtf drivers." Of course, the later statement "except in cases where they suppress other wtf drivers" itself is a function of the antidote, not the driver, so it does not contribute to the maintenance of the drive.

While the authors seem to dismiss the possibility of another function of wtf drivers, such a function seems necessary for their model (although, this is somewhat unclear as the model is not fully fleshed out), or perhaps losing the poison function while retaining the antidote is an unlikely mutation? Anyways, this seems to require more thought, and I believe it is a mistake to write this off as "we haven't found evidence of this" as it is not clear the author's model works without some additional benefit.

*Reviewer #3 (Recommendations for the authors):*

Results:

Given the high divergence, I wonder if these genes are better described as wtf-like genes rather than wtfs.

Figure 2 Figure supplement 1 is not very informative. Are these all wtf-related sequences from all species? Just the antidote regions? Please revise the legend to help the reader understand what is being plotted.

Figure 2—figure supplement 3: Is the x-axis un-normalized read counts? It seems some normalization for length would be appropriate here.

The low contrast in figure 1A makes it difficult to read. I suggest increasing the contrast (e.g. using a lighter background) and ideally avoiding the combination of red and green to make it color-blind friendly.

Pg 10 line 261: Please be quantitative.

Pg 7 line 186: say that these are ONT reads here.

It's better to use quantitative language on pg 9 line 218 regarding Figure 2A.

It would be helpful to mention in the text where the 3E-6 P-value comes from, and what you mean by the comparison with Mei4 targets. I had to read the methods a few times to understand what you did with this analysis.

I'm not sure how to interpret the cytological patterns you see when you express the non-pombe wtfs in *S. cerevisiae*. What is the significance of the differences between the aggregate appearance relative the vacuole etc.? You are trying to learn something about functional conservation but you are just looking at protein localization, and it is outside of its natural context (e.g. S. octosporus). I'm not sure how that relates to the protein's function. If the localization patterns were the same as *S. pombe*, I'm not sure that you could conclude that there is functional conservation. The patterns are worth looking at, and the co-localization of poison and antidote is interesting and informative, but the discussion in the context of 'function' is unclear.

Pg 24 Line 614-615 I had to read the line several times. Rephrase for clarity?

In the methods, the description of the mei4 gene discovery was unclear to me. There might be some jargon that needs more explanation. For example line 920 ("middle meiosis genes"). Also, the analysis described on line 933 was unclear to me in the text and in the methods.

Discussion:

Pg 26 Line 667: Rsp-like sequences are related to the targets of drive but are not known to be targets or associated with drive themselves.

Pg 26 line 671-672: What is the evidence that Tf LTRs and 5S rDNA genes cluster spatially in the nucleus?

Pg 27 line 679: This sentence is unclear. What does 'Factors found in the clusters" mean?

Pg 27 lines 684-685: I'm not sure what you mean by "spread to distributed repetitive sequences that cluster within nuclei".

Pg 27 lines 685-686: I don't think that these are 359 repeats. They are related to sequences in the 1.688 family (which includes the 359-bp satellite).

Pg 27 line 693: What does "single, stationary drive locus" mean?

One might argue that wtfs drive systems are simple in that they only involve one gene family. Your point may be clearer without using the word "complexity" to describe the feature driving the birth-death cycle.

Page 27 line 699: "Short-circuiting the two main paths to extinction" maybe be unclear to some readers. Consider simplifying your prose.

Pg 28 line 710. What do you mean here? You just told us that wtf suppressors are specific to drivers.

Pg 28 lines 713-714: This language is confusing.

Pg 29 lines 741-747. I'm not sure about the motivation of this paragraph.

Comment about the organization: It would be very helpful to your readers and evaluators if figures appeared with their legends and ideally near where they are cited in the text.

---

## [Author Response]

Essential revisions:1) To ensure your conclusions are bounded by the results, replace your verbal model with a discussion of newly discovered mechanisms of wtf evolution (e.g., expansion, gene conversion). In this discussion, take care to acknowledge that questions remain about the timescale of wtf fixation relative to the emergence of new drivers.

We significantly changed the discussion and formally acknowledged gaps in our understanding including: (1) timescale about driver fixation relative to rebirth and (2) our limited understanding of *S. pombe* ecology. We also clarified our writing to distinguish fixation within a local population from fixation within a species. Finally, we removed the model figure (Figure 11).

2) Please temper your claims in the Introduction and Discussion sections related to finding evidence of ACTIVE drive across the evolutionary history of this group.

We changed our wording to clarify that the gene family has retained the ability to drive, but that we have not demonstrated perpetual drive over the last 100 million years.

3) Temper your claims about protein localization patterns and functional conservation.

We changed the text to make the logic of the analyses in *S. cerevisiae* clear. We also changed the text to clarify that our claims for functional conservation were based on several lines of evidence, including protein localization data.

4) Adjust your wording to avoid seeming dismissive of additional (non-drive) functions for wtf genes. Also, rewrite to clarify their role as suppressors.

We changed this paragraph to point out that the critical experiments for assessing non-

drive functions have not yet been reported. We also changed the wording to be less

dismissive that such work may identify beneficial roles for wtf genes.

We also clarified that wtf drivers can be suppressed by genes that encode only

antidotes, and by other drivers if they encode a highly similar antidote.

Reviewer #2 (Recommendations for the authors):The "model".My sense is the paper would be improved by replacing the "model" with a brief sentence about, the diverse evolutionary mechanisms discovered here, which likely play a key role in the existence of wtf-based drive in the long-diverged taxa found here.

We did not fully remove discussion of how the molecular mechanisms and evolutionary patterns of the wtf genes may could be contributing to their longevity. We have, however, used the reviewer comments to improve and clarify our points.

Does wtf somehow otherwise impact fitness.The authors present and disregard an additional potential part of the evolution of wtf drivers… "… [wtf-driver] genes are not merely selfish parasites… [but may] promote fitness in some way that [they] have yet to discover. This theoretical additional function of the wtf genes could have promoted their long-term maintenance in fission yeast genomes. It is important to note, however, that genes do not need to promote fitness to be maintained in genomes and there is currently no evidence supporting a role of wtf drivers in promoting fitness, except in cases where they suppress other wtf drivers." Of course, the later statement "except in cases where they suppress other wtf drivers" itself is a function of the antidote, not the driver, so it does not contribute to the maintenance of the drive.

We also clarified the more the mechanism of Wtf protein function and routes to suppression. Importantly, each *wtf* driver encodes both a poison and an antidote. We have found that drivers can suppress each other if they share highly similar sequences.

While the authors seem to dismiss the possibility of another function of wtf drivers, such a function seems necessary for their model (although, this is somewhat unclear as the model is not fully fleshed out), or perhaps losing the poison function while retaining the antidote is an unlikely mutation? Anyways, this seems to require more thought, and I believe it is a mistake to write this off as "we haven't found evidence of this" as it is not clear the author's model works without some additional benefit.

The loss of poison function is not an unlikely mutation. The dedicated suppressors of

drive we describe in the text are genes that encode only antidote proteins. Additionally,

we have rewritten this paragraph to be less dismissive of potential additional roles of wtf genes.

Reviewer #3 (Recommendations for the authors):Results:Given the high divergence, I wonder if these genes are better described as wtf-like genes rather than wtfs.

Given the homology, functional conservation, and its inherent charms, we decided to

retain the wtf name.

Figure 2 Figure supplement 1 is not very informative. Are these all wtf-related sequences from all species? Just the antidote regions? Please revise the legend to help the reader understand what is being plotted.

We have revised this legend to clarify.

Figure 2—figure supplement 3: Is the x-axis un-normalized read counts? It seems some normalization for length would be appropriate here.

These are long-read RNA seq data and the X-axis read counts are not normalized. The genes considered in the plot do not vary greatly in size.

The low contrast in figure 1A makes it difficult to read. I suggest increasing the contrast (e.g. using a lighter background) and ideally avoiding the combination of red and green to make it color-blind friendly.

We have changed the figure as suggested.

Pg 10 line 261: Please be quantitative.

We have revised to indicate the fraction of genes with the motif.

Pg 7 line 186: say that these are ONT reads here.

We have added this information.

It's better to use quantitative language on pg 9 line 218 regarding Figure 2A.

We added a new figure (Figure 2-source data 2) showing the average exon and intron sizes of the *wtf* genes from all species.

It would be helpful to mention in the text where the 3E-6 P-value comes from, and what you mean by the comparison with Mei4 targets. I had to read the methods a few times to understand what you did with this analysis.

We reworded this section to better explain the logic behind our P value cutoff choice.

I'm not sure how to interpret the cytological patterns you see when you express the non-pombe wtfs in S. cerevisiae. What is the significance of the differences between the aggregate appearance relative the vacuole etc.? You are trying to learn something about functional conservation but you are just looking at protein localization, and it is outside of its natural context (e.g. S. octosporus). I'm not sure how that relates to the protein's function. If the localization patterns were the same as *S. pombe*, I'm not sure that you could conclude that there is functional conservation. The patterns are worth looking at, and the co-localization of poison and antidote is interesting and informative, but the discussion in the context of 'function' is unclear.

We have revised this section extensively to clarify. In short, the general mechanism of Wtf proteins is that the poison co-assembles with the antidote and promotes it’s trafficking to the vacuole using a broadly conserved pathway. That was first observed with *wtf4* from *S. pombe*, but the same thing happens with the proteins encoded by *S. octosporus wtf25.*

Pg 24 Line 614-615 I had to read the line several times. Rephrase for clarity?

We have revised this to clarify.

In the methods, the description of the mei4 gene discovery was unclear to me. There might be some jargon that needs more explanation. For example line 920 ("middle meiosis genes"). Also, the analysis described on line 933 was unclear to me in the text and in the methods.

Thanks for pointing this out. We reworded this section in the results and the methods.

Discussion:Pg 26 Line 667: Rsp-like sequences are related to the targets of drive but are not known to be targets or associated with drive themselves.

Thank you for pointing this out. We have reworded to correct this error.

Pg 26 line 671-672: What is the evidence that Tf LTRs and 5S rDNA genes cluster spatially in the nucleus?

We added references to support these statements.

Pg 27 line 679: This sentence is unclear. What does 'Factors found in the clusters" mean?

We revised this section to clarify.

Pg 27 lines 684-685: I'm not sure what you mean by "spread to distributed repetitive sequences that cluster within nuclei".

We have revised this to clarify.

Pg 27 lines 685-686: I don't think that these are 359 repeats. They are related to sequences in the 1.688 family (which includes the 359-bp satellite).

We double checked and both works referenced referred to them as 359.

Pg 27 line 693: What does "single, stationary drive locus" mean?

This sentence has been removed.

One might argue that wtfs drive systems are simple in that they only involve one gene family. Your point may be clearer without using the word "complexity" to describe the feature driving the birth-death cycle.

We reworded this section.

Page 27 line 699: "Short-circuiting the two main paths to extinction" maybe be unclear to some readers. Consider simplifying your prose.

We reworded this to be more precise.

Pg 28 line 710. What do you mean here? You just told us that wtf suppressors are specific to drivers.

We were referring to potential non-*wtf* suppressors, but have removed this section.

Pg 28 lines 713-714: This language is confusing.

This section has been removed.

Pg 29 lines 741-747. I'm not sure about the motivation of this paragraph.

This section has been removed.